

# Using Empirical Orthogonal Teleconnections to evaluate interannual rainfall variability over China in the Met Office Unified Model Global Atmosphere 6.0 and Global Coupled 2.0 configurations

Claudia Christine Stephan[1], Nicholas P Klingaman[1], Pier Luigi Vidale[1], Andrew G Turner[1,2],
Marie-Estelle Demory[1,3], and Liang Guo[1]

[1]National Centre for Atmospheric Science – Climate, Department of Meteorology, University of Reading, P.O. Box 243, Reading RG6 6BB, United Kingdom
[2]Department of Meteorology, University of Reading, P.O. Box 243, Reading RG6 6BB, United Kingdom
[3]Center for Space and Habitability, University of Bern, Gesellschaftsstrasse 6, 3012 Bern, Switzerland

*Correspondence to:* Claudia Stephan (c.c.stephan@reading.ac.uk)

**Abstract.** Six climate simulations of the Met Office Unified Model Global Atmosphere 6.0 and Global Coupled 2.0 configurations are evaluated against observations and reanalysis data for their ability to simulate the mean state and year-to-year variability of precipitation over China. To analyze the sensitivity to air-sea coupling and horizontal resolution, atmosphere-only and coupled integrations at atmospheric horizontal resolutions of N96, N216 and N512 (corresponding to ∼200, 90, and

5  40 km in the zonal direction at the equator, respectively) are analyzed. The mean and interannual variance of seasonal precipitation are too high in all simulations over China, but improve with finer resolution and coupling. Empirical Orthogonal Teleconnection (EOT) analysis is applied to simulated and observed precipitation to identify spatial patterns of temporally coherent interannual variability in seasonal precipitation. To connect these patterns to large-scale atmospheric and coupled air-sea processes, atmospheric and oceanic fields are regressed onto the corresponding seasonal-mean timeseries. All simulations

10  reproduce the observed leading pattern of interannual rainfall variability in winter, spring and autumn; the leading pattern in summer is present in all but one simulation. However, only in two simulations are the four leading patterns associated with the observed physical mechanisms. Coupled simulations capture more observed patterns of variability and associate more of them with the correct physical mechanism, compared to atmosphere-only simulations at the same resolution. However, finer resolution does not improve the fidelity of these patterns or their associated mechanisms. This shows that evaluating climate

15  models by only geographical distribution of mean precipitation and its interannual variance is insufficient. The EOT analysis adds knowledge about coherent variability and associated mechanisms.

*Copyright statement.* TEXT





# 1 Introduction

Rainfall over China is mainly influenced by the East Asian Summer Monsoon (EASM), characterized by southerly winds, and the East Asian Winter Monsoon (EAWM), featuring northerly winds. Annual mean precipitation decreases from southeast to northwest China. Precipitation exhibits a strong seasonal cycle, as well as substantial intraseasonal, interannual and decadal
variability. Droughts, floods and cold surges severely impact the lives of millions of people by affecting water resource management, infrastructure and the ecological environment (Huang et al., 1998, 1999; Huang and Zhou, 2002; Barriopedro et al., 2012; Huang et al., 2012b). Therefore, regional interannual variability (IAV) of rainfall is likely to cause more damage to human life, agriculture and infrastructure than a slowly changing mean state arising from climate change.

A complex mixture of physical mechanisms and a highly spatially variable climate and orography make it difficult for
General Circulation Models (GCMs) to correctly reproduce the geographical distribution of precipitation and its variability over China. Precipitation is influenced by local air-sea interactions over the South China Sea (SCS), by tropical and subtropical circulation systems (Ding, 1994; Zhou et al., 2011), and by teleconnections to modes of variability in the Indian and Pacific Oceans.

The modulation by the El Niño Southern Oscillation (ENSO) has been studied for many years. Developing El Niños fa-
vor droughts in northern China (Huang and Zhou, 2002). A weaker EAWM has been associated with El Niño, and a stronger monsoon with La Niña (Zhang et al., 1996; Wang et al., 2000). Summer flooding in the Yangtze River valley can occur during the decaying stage of El Niño (Wang et al., 2001; Zhang et al., 2007; Xie et al., 2016). In southern China, following El Niño, spring rainfall increases (Zhang et al., 2016), while autumn rainfall is reduced (Zhang et al., 2013; Wang and Wang, 2013). Extratropical teleconnection patterns also affect interannual rainfall variability over East Asia (Chen and Huang, 2014; Lu et al.,
2002; Tao et al., 2016; Wang and Feng, 2011).

During recent decades GCMs have become more complex by coupling to dynamical ocean models and including more, and more complex, sub-grid scale parameterizations. GCM assessment efforts, such as the third and fifth Coupled Model Intercomparison Projects (CMIP3 and CMIP5, Meehl et al., 2007; Taylor et al., 2012), show that large biases remain in the simulation of the East Asian monsoon (Sperber et al., 2013). Over China most current GCMs produce cold near-surface temperature biases
and excessive precipitation. In addition, GCMs underestimate the southeast-northwest precipitation gradient and overestimate the magnitude and spatial variability of IAV, with little change from CMIP3 to CMIP5 (Jiang et al., 2005, 2016).

Increases in computational power allow GCMs to run at increasingly finer horizontal resolution. Previous studies have investigated the effect of resolution on the fidelity of the simulation of rainfall and IAV in China. Jiang et al. (2016) grouped coupled GCMs participating in CMIP3 and CMIP5 into groups of $< 2°$ (222 km), $2°$–$3°$ (222–334 km) and $> 3°$ (334 km)
resolution and found that finer resolution improved GCM fidelity for winter, spring, and autumn precipitation over China. Jiang et al. (2016) found no resolution sensitivity for precipitation in summer. This is consistent with Song and Zhou (2014), who examined the simulation of the EASM in CMIP3 and CMIP5 atmosphere-only GCMs and found no evidence for a relationship between model fidelity and resolution (ranging from ∼22–500 km). In their analysis of CMIP5 coupled models, Chen and Frauenfeld (2014) also concluded that horizontal resolution did not influence climatological precipitation and IAV





over China. On the other hand, Li et al. (2015) performed a series of experiments with the Community Atmosphere Model (CAM5) at resolutions of T42 (310 km at the equator), T106 (125 km), and T266 (50 km). At finer resolution, spatial patterns of rainfall along the southern edge of the Himalayas became more realistic, and a large positive precipitation bias to the east of the Tibetan Plateau reduced significantly, due to more realistic orography. Based on experiments with a regional climate model

at resolutions of 45–240 km and keeping all other settings identical, Gao et al. (2006) also found that horizontal resolution of 60 km or finer was important to accurately simulate monthly mean precipitation over China. By comparing to a set of simulations with variable resolution along with degraded-resolution topography, they concluded that the improvement was due to better resolved physical processes rather than a better resolved topography.

In contrast, few studies have directly assessed the effects of air-sea coupling on the fidelity of East Asian precipitation. In

experiments with coupled and atmosphere-only GCMs, Misra (2008) found that including coupling is important for correctly simulating the long auto-decorrelation time of daily rainfall in summer monsoon regions. However, coupled models struggle to correctly represent the seasonal cycle of SSTs and the spatial pattern and magnitude of ENSO SST anomalies (Latif et al., 2001; Bellenger et al., 2014). Errors in position or amplitude of equatorial ENSO SSTs have substantial impacts on teleconnections to rainfall in CMIP1/2 (AchutaRao and Sperber, 2002) and CMIP3 (Cai et al., 2009) models. Gong et al. (2014) found that the

spatial structure and amplitude of ENSO-related SST anomalies are key factors for the ENSO-EAWM relationship in CMIP5 models. Even in atmosphere-only models and in the absence of SST biases, observed rainfall patterns associated with ENSO are poorly captured, with little or no improvement from CMIP3 to CMIP5 (Langenbrunner and Neelin, 2012).

Improvements in GCMs rely on continuous evaluation at the regional scale, as well as understanding the physical processes that must be captured to reliably simulate regional climate and its variability. The latter is particularly important for a densely

populated country such as China. Previous studies have mainly assessed the ability of GCMs to simulate the distribution of mean precipitation and its interannual standard deviation over China. This study adds a comprehensive assessment of the leading patterns of coherent precipitation variability in all seasons, using Empirical Orthogonal Teleconnection (EOT) analysis. EOT analysis objectively identifies regions in China that show strong coherent IAV in seasonal precipitation. Rotstayn et al. (2010) showed that EOT analysis is useful for assessing regional climate variability in GCMs through analysis of the leading

patterns of Australian annual rainfall. In a previous study, Stephan et al. (2017a) performed EOT analysis on 1951–2007 high-resolution gridded precipitation data over China in all seasons. Their results serve as a benchmark for our model assessment.

Changes in the physics schemes between different models participating in CMIP experiments may obscure the direct effects of changes in resolution or the addition of air-sea coupling. Therefore, we analyze simulations with Met Office Unified Model (MetUM) Global Atmosphere 6.0 and Global Coupled 2.0 configurations at resolutions of N96, N216 and N512 (corresponding

to ∼200, 90, and 40 km in the zonal direction at the equator); physical parametrizations remain unchanged. This allows us to focus on the effects of coupling and resolution. We aim to answer three questions: (i) do simulations produce observed patterns of regional precipitation variability? (ii) are those patterns associated with the observed physical mechanisms? (iii) is the fidelity of these patterns and their associated mechanisms sensitive to horizontal resolution and/or air-sea coupling?





Section 2 describes the model simulations, observational data sets and analysis methods. Section 3 presents model biases in seasonal mean precipitation and IAV; Sect. 4 reports biases in the teleconnection to ENSO. Results from the EOT analysis are shown in Sect. 5. Section 6 is a discussion; Sect. 7 summarizes the main results.

## 2 Data and methods

### 5 2.1 MetUM simulations

We analyze two AMIP-style simulations from 1982–2008 at resolutions of N96 ($1.875° \times 1.25°$, 208 km $\times$139 km in longitude and latitude at the equator) and N216 ($0.83° \times 0.55°$, 93 km $\times$62 km). We refer to these as A96 and A216, respectively. They use the Global Atmosphere configuration 6.0 of the MetUM (GA6, Walters et al., 2017), driven by monthly mean sea surface temperatures (SSTs) from the Reynolds product (Reynolds et al., 2007) and varying solar, greenhouse gas and aerosol forcings

as observed in 1982–2008.

The coupled simulations use the Global Coupled configuration 2.0 of the MetUM (GC2, Williams et al., 2015). We analyze two GC2 simulations at N96 and N216 resolutions in the atmosphere, referred to as C96 and C216, that are initialized with present-day ocean data (EN3; Ingleby and Huddleston, 2007) and spun-up sea-ice and land surface conditions. We further analyze two coupled simulations (C512a and C512b) at N512 resolutions ($0.35° \times 0.23°$, 39 km $\times$26 km). Both are initialized

with ocean conditions from a previous N512 simulation, but with an offset of 55 years to sample different phases of decadal variability. The coupling frequency is 3 hours and the ocean resolution is fixed at $0.25°$ on the ORCA025 tri-polar grid (Madec, 2008).

The four GC2 simulations are present-day control simulations using emissions and solar forcing with constant 1990 values and are integrated over 100 years. Observed 1961–2012 precipitation trends over China do not exceed 2 mm year$^{-1}$ in any

season in any region (Wang and Yang, 2017). The interannual variability of the EOT timeseries are typically two orders of magnitude larger. Therefore, differences in the applied forcing between the GA6 and GC2 simulations can be assumed to have negligible effects on the results. Table 1 summarizes the simulations. Figure 1 shows the grid-point mean orographic height at each resolution. As described in Walters et al. (2017), subgrid orographic boundary layer drag, flow blocking and gravity wave drag are parameterized in the MetUM. With finer resolution, more orography is resolved explicitly, and less is parameterized.

### 25 2.2 Empirical Orthogonal Teleconnections

In a previous study, Stephan et al. (2017a) performed EOT analysis on 1951–2007 gridded APHRODITE precipitation data (see Sect. 2.3). Unlike Empirical Orthogonal Functions (EOFs), which are orthogonal in space and time, EOTs are orthogonal either in space or time (Van den Dool et al., 2000). To investigate coherent temporal rainfall variability, we first compute the China-averaged timeseries of rainfall anomalies and search for the grid point that best explains its variance (Smith, 2004). All

MetUM precipitation data are interpolated to the APHRODITE grid to allow for a comparison with observations. We chose the APHRODITE grid so that the observed EOT patterns do not change from the ones reported in Stephan et al. (2017a).





To quantify the spatial coherence of rainfall we correlate rainfall anomalies at each grid point with this so-called base point. To compute EOTs of second and lower orders, we remove all higher-order EOT timeseries from all points in the domain by linear regression and then repeat the above steps. We compute up to three EOTs because lower-order patterns explain only small amounts of variance. The EOT method identifies independent regional patterns of coherent rainfall variability. The associated timeseries can be used to connect the patterns to regional and large-scale atmospheric and coupled atmosphere-ocean mechanisms, as Stephan et al. (2017a, b) demonstrated for China, and as was previously shown for other regions (Smith, 2004; Rotstayn et al., 2010; Klingaman et al., 2013; King et al., 2014).

It is important to note that EOT analysis does not maximize the percentage of explained space-time variance, resulting in several benefits: i) the associated timeseries are connected to specific grid points, ii) the lack of spatial orthogonality means it is possible to find more than one EOT that explains variance in the same area, iii) because EOTs are orthogonal only in time, the method is less sensitive to the choice of the spatial domain.

Connections between rainfall patterns and drivers are established by linearly regressing atmospheric and oceanic fields onto timeseries corresponding to each EOT pattern. We show regressions associated with a one standard deviation increase in an EOT timeseries, but all discussions hold for negative anomalies in the EOT timeseries, with the opposite sign. We use Spearman's rank correlations because the rainfall data are not normally distributed.

To test whether MetUM simulations reproduce observed patterns, we interpolate simulated precipitation to the APHRODITE grid, and regress China-wide seasonal precipitation against simulated and observed EOT timeseries. If the linear Pearson pattern correlation coefficients between simulated and observed regression maps are greater than a subjectively chosen threshold of 0.38, we consider them a match. This value was chosen because patterns with a smaller correlation were located far away from the respective observed pattern.

## 2.3 Observational data and indices

Seasonal-total precipitation for winter (DJF), spring (MAM), summer (JJA) and autumn (SON) over China is obtained from the Asian Precipitation - Highly-Resolved Observational Data Integration Towards Evaluation (APHRODITE) data set (Yatagai et al., 2012). It is a long-term (1951–2007), continental-scale daily product with a resolution of $0.5° \times 0.5°$, produced from rain-gauge data. Before the data are gridded, an objective quality control procedure is applied (Hamada et al., 2011). Previous studies confirmed the high quality of the data (Rajeevan and Bhate, 2009; Krishnamurti et al., 2009; Wu and Gao, 2013). The rain gauge network in China is dense except in climatologically dry regions, i.e., northwest China and the Tibetan Plateau [see Yatagai et al. (2012) for a map of rain-gauge locations]. However, EOT patterns tend to peak in climatologically wet regions, and all EOTs in this study are in areas of high-density observations.

We show regressions onto the EOT timeseries of surface pressure ($P_{SFC}$), 200 hPa geopotential height ($Z_{200}$) and 850 hPa wind from the European Centre for Medium-Range Weather Forecasting Interim global reanalysis (ERA-Interim, from now on also referred to as observations), available at $0.7° \times 0.7°$ resolution from 1979–2007 (Dee et al., 2011). We compute area averages of $P_{SFC}$ in the Northwest Pacific (NWP, 10–30° N, 120–170° E) and 500 hPa geopotential height ($Z_{500}$) in 25–45° N, 120–140° E, to connect patterns of rainfall variability with circulation anomalies in the NWP and around Japan, respectively.





SSTs for 1870–2010 are obtained from the $1° \times 1°$ Hadley Centre sea Ice and SST data set (HadISST; Rayner et al., 2003). We average linearly detrended SSTs in the South China Sea (SCS, 5–25° N, 100–120° E) and in the eastern tropical Pacific (Niño3.4, 5° S–5° N, 190–240° E) to associate rainfall anomalies with ocean variability.

As a proxy for convective activity we show regressions onto EOT timeseries of seasonal mean $2.5° \times 2.5°$ interpolated
satellite-retrieved outgoing longwave radiation data (OLR; Liebmann and Smith, 1996), available for 1974–2013.

## 3  Climatology

MetUM GA6 and GC2 show substantial biases in seasonal mean precipitation and IAV (Fig. 2). Observed DJF mean precipitation does not exceed 300 mm anywhere in China, but all simulations show substantially larger amounts. A96 produces wet biases of ∼100 mm and ∼50–100 % larger IAV in southwest and southeast China. Increasing resolution from N96 to N216 in
coupled (C216 vs. C96) and atmosphere-only (A216 vs. A96) simulations improves biases near orography in southwest China (compare Fig. 1). Positive biases in seasonal mean precipitation and IAV in the N512 simulations are confined to southeast China. Observed IAV is largest along the southeast coast, but in A96 and A216 it peaks further north along the eastern Yangtze basin. In this area, higher resolution increases the positive bias in IAV by ∼50 %, as is also seen between C96 and C216. This may be related to orographically-forced precipitation (Li et al., 2015). However, adding coupling and resolution tends to shift
areas of high IAV further south, closer to where high variability is observed. C512a and C512b still overestimate observed variability, but regions of high variability agree better with observations, i.e. they are located in the southeast of China and south of the Yangtze valley.

The positive precipitation bias in MetUM GA6/GC2 is consistent with a weaker than observed EAWM circulation (not shown). Because the EAWM circulation is strongest in the lower troposphere over coastal East Asia (Ding, 1994), we compute
the spatial correlation coefficient (r) and the normalized root-mean square error (e) of simulated relative to observed meridional winds at 10 m in (25°–40° N, 120°–140° E) plus (10°–25° N, 110°–130° E), as in Jiang et al. (2016). Increasing resolution from N96 to N216 improves the spatial correlation and reduces the error (r=0.75, e=0.84 in A96 versus r=0.93, e=0.59 in A216; r=0.64, e=1.02 in C96 versus r=0.82, e=0.82 in C216). No improvement is seen from C216 to C512. The most drastic improvement occurs in the SCS where there are strong southerly biases in N96 of over 4 ms$^{-1}$. These southerly wind biases
are greater in the coupled simulations due to a stronger western Pacific subtropical high.

In MAM the spatial patterns of seasonal mean precipitation and IAV are similar in observations and simulations. All simulations show positive biases in mean precipitation and IAV, particularly in southeast China south of the Yangtze River, where the bias in A96 is >400 mm, double the observed precipitation; variability is also doubled. Increasing resolution to N216 reduces wet biases in southwest China in the GA6 and GC2. Coupling at N96 and N216 reduces wet biases in southeast China;
however, there is little change between C216 and C512.

In JJA climatological observed rainfall decreases from southeast to northwest China; variability is highest in the eastern Yangtze valley. In A96 total rainfall is too high and the isolines are too zonal, creating large wet biases in southwest China and the Tibetan Plateau, consistent with the poor simulation of the southeast-northwest precipitation gradient in CMIP3 and





CMIP5 (Jiang et al., 2016). IAV is doubled compared to observations in south China and the Yangtze valley. In most areas biases become small with finer resolution and coupling; C216 and C512 compare well with observations except for a wet bias in south China. To measure the strength of the EASM circulation we again follow Jiang et al. (2016) and compare 850 hPa meridional winds in 20°–40° N, 105°–120° E. The EASM is too weak in all simulations, which results in a weak northward

transport of moisture and positive precipitation biases in south China (not shown).

In SON observed total rainfall in southeast China is ∼200–300 mm, and rainfall variability is relatively spatially homogenous compared to the other seasons. In A96 total precipitation and its variability are realistic in the eastern part of southeast China, but wet biases become large toward the Tibetan Plateau. Biases reduce with resolution and coupling.

Comparing GA6 and observations using their common record period does not change the above findings. In summary, mean

precipitation and IAV tend to be too high in the MetUM. Finer resolution reduces biases near steep topography, particularly in southwest China. Adding coupling reduces wet biases along the eastern Yangtze valley in DJF and JJA, and in southeast China in MAM.

## 4  Teleconnection to ENSO

Since ENSO has an important influence on IAV in China, particularly in DJF, MAM and JJA (see Sect. 1), we now examine

how well the MetUM captures climatological SSTs and the teleconnection to ENSO. The annual mean SST bias in the coupled simulations (Fig. 3) is characterized by cold biases in the northern hemisphere and a warm bias in the Southern Ocean. These biases are associated with an Inter-Tropical Convergence Zone (ITCZ) that is displaced to the south over the Atlantic and Indian Ocean (Williams et al., 2015). In C512 the Indian Ocean has a slight warm bias, when in C96 and C216 it has a slight cold bias. Compared to C96, C216 and observations, SSTs in the tropical Pacific and tropical Atlantic are also warmest in C512

(Fig. 3). GC2 compares favorably with other CMIP5 models (Bellenger et al., 2014), simulating an approximately correct spatial pattern of ENSO SSTs, albeit with slightly weaker variability in the central Pacific, a power spectrum with frequencies within the observed range (3–7 years), and good seasonality with maximum (minimum) variability in boreal winter (spring) (Williams et al., 2015). For our four GC2 simulations specifically, the ratios of simulated to observed standard deviations of DJF SSTs in the Niño3.4 region are 0.61 (C96), 0.71 (C216, C512a) and 0.83 (C512b).

In observations, ENSO is positively correlated with DJF precipitation in southeast China (Fig. 4). A96 produces enhanced rainfall in southeast China in DJF in response to ENSO, but the maximum is shifted to the eastern Yangtze valley and southwest China is too dry. A216 produces only a slightly better agreement with observations. C96 has no significant teleconnection to ENSO; C216 and C512 on the other hand compare well to observations.

The observed rainfall pattern in MAM consists of anomalously dry areas in southeast China and anomalously wet areas

in eastern China in response to ENSO. All simulations reproduce a dipole pattern, but statistically significantly increased precipitation to the north of the Yangtze valley is only seen in C216 and C512.

DJF Niño3.4 is positively correlated with precipitation in the following JJA along the central Yangtze valley. A96 shows increased precipitation along the Yangtze River, but also in southeast China. In A216 precipitation is also increased in southeast





China, and reduced in southwest China and between the Yangtze and Huaihe Rivers. All coupled simulations show a very weak teleconnection to ENSO. Only in C96 are there positive anomalies in the Yangtze valley. Observed JJA circulation anomalies following DJF ENSO are characterized by a strong anticyclonic circulation in the western North Pacific (Fig. 5). Dynamical mechanisms proposed for this anticyclonic circulation include suppressed convection over the western equatorial Pacific due

to a weakened Walker circulation during El Niño (Zhang et al., 1996), Rossby waves triggered by SST anomalies over the western tropical Pacific (Matsuno, 1966; Gill, 1980; Wang et al., 2000), and the nonlinear atmospheric interaction between the annual cycle and ENSO variability (Stuecker et al., 2013, 2015; Xie and Zhou, 2017). Explanations for the long persistence include local wind-evaporation-SST feedback (Wang et al., 2000) and the delayed Indian Ocean warming (e.g., Yang et al., 2007; Xie et al., 2009). A96 and A216 produce northeasterlies over southeast China instead of southwesterlies (Fig. 5), which

may be associated with a weak western North Pacific subtropical high that does not extend far enough westward (not shown). The pattern of the circulation response in GC2 is similar at all resolutions and scales with the strength of ENSO variability (Fig. 4). Over southeast China there are weak southwesterlies in C96 and C216. In C512 westerlies are present over the eastern Yangtze-Huaihe basin. Pacific wind anomalies in GC2 are more zonal compared to GA6, and compare better to observations. Hence, the JJA response to ENSO is sensitive to coupling, but no MetUM configuration produces the observed response.

Observed rainfall anomalies in SON show a dipole of wet conditions in southern China and dry conditions in central China. The simulations show no consistent response to ENSO. However, the observed SON EOTs in this study are not related to ENSO.

Coupling and higher resolution improve the teleconnection to ENSO in DJF and in MAM, but the teleconnection is poor in JJA in all simulations.

## 20  5   EOT results

Based on APHRODITE data, Stephan et al. (2017a) identified the dominant regions of observed coherent IAV in seasonal precipitation in China, and linked that variability to large-scale and regional mechanisms. They analyzed only EOTs that explained at least 5 % of the total space-time variance, which resulted in two patterns in DJF, JJA, SON, and three in MAM. They found connections to ENSO for DJF precipitation variability in large parts of eastern China (DJF Obs-1), MAM precipitation vari-

ability in southeast China (MAM Obs-1), and JJA precipitation variability in the southern Yangtze River valley (JJA Obs-1). Relationships to extratropical wave propagation were found for DJF along the southeast coast of China (DJF Obs-2), MAM in the Yangtze region (MAM Obs-2) and the northern parts of eastern China (MAM Obs-3), and for large areas of eastern China in SON (SON Obs-1). JJA IAV in the northern areas of the Yangtze region (JJA Obs-2) and SON IAV in the coastal area of southern China (SON Obs-2) were associated with local pressure anomalies.

Here, we compare simulated and observed EOTs for each season in turn. Results based on APHRODITE observations will be denoted 'Obs-1', 'Obs-2', 'Obs-3', and results based on model simulations will be denoted by the simulation identifier and the order of the EOT (e.g., 'A96-2' stands for the second-order EOT in A96).

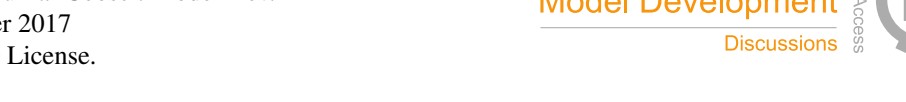

### 5.1 Winter

#### 5.1.1 Pattern 1

Obs-1 describes rainfall variability in large areas of eastern China (Fig. 6) and explains 34 % of the space-time variance. It is associated with ENSO, with a correlation coefficient of 0.44 with DJF Niño3.4 (Table 2). The Walker circulation is weakened,

with negative $P_{SFC}$ anomalies in the eastern tropical Pacific and positive anomalies in the western tropical Pacific (r=0.42 with the NWP $P_{SFC}$ index, Table 2). The EAWM is weakened, and anomalous southwesterlies along the coast of southeast China transport moisture from the Bay of Bengal and the SCS (not shown), where SSTs are anomalously warm (r=0.40 with SCS SST index, Table 2).

All simulations reproduce the leading spatial pattern (Fig. 6), with pattern correlation coefficients between 0.85 (A96) and

0.94 (C96, C512a), and similar percentages of explained variance (Table 2).

The C216 and C512 EOT timeseries are statistically significantly correlated with Niño3.4, SCS SST and NWP $P_{SFC}$, with correlation coefficients similar to observations (Table 2). Global regressions of SST and $P_{SFC}$ on DJF EOT 1 confirm ENSO signals (not shown). C96, A96 and A216 miss the connection to ENSO. This result is expected, because only C216 and C512 have a realistic teleconnection to ENSO in DJF (Sect. 4). Note that for observations, the correlation to Niño3.4 is only

0.44, so ENSO explains only ∼40 % of the variance in the EOT timeseries. There are positive $Z_{200}$ anomalies over Europe, northern Africa and northeast Asia, and negative anomalies over the Middle East and southern China in ERA-Interim. Similar extratropical $Z_{200}$ perturbations exist in all simulations. Such perturbations can arise from a variety of pathways and are not always connected to ENSO.

#### 5.1.2 Pattern 2

Obs-2 peaks along the southeast coast with another area of coherent rainfall variability further north along the coast. It explains 12 % of the space-time variance, and is associated with low $Z_{200}$ over China, positive $P_{SFC}$ over and to the east of Japan (fourth row in Fig. 7), negative OLR anomalies over the western equatorial Indian Ocean and positive OLR anomalies over the eastern equatorial Atlantic Ocean (not shown). Stephan et al. (2017a) showed that the pressure distribution is consistent with a wavenumber-3 Rossby wave originating over central Africa or the Arabian Sea and propagating across China and Japan into

the Pacific.

A pattern with a peak along the southeast coast is found in A216, C96, C216 and C512b. Only A96 produces a coastal pattern with separate southern and northern areas of covariability. The A96 pattern has a correlation of 0.53 with Obs-2, less than the other simulations [between r=0.70 (C96) and r=0.83 (C216)]. Only in A96 is the base point situated inside the northern area. The pattern describes between 8 % (A216) and 14 % (C512b) of the space-time variance in the simulations.

In A216, C216 and C512b the pattern is weakly correlated with ENSO (Table 2). In C96 northward moisture transport is associated with low pressure over South Asia (Fig. 7). In A96 onshore flow is created by high pressure over the Korean peninsula (Fig. 7). We conclude that the simulations that capture Obs-2 do not associate precipitation with the correct large-scale driving mechanism. The pressure anomalies associated with Obs-2 are not present in any of the simulations.



### 5.2 Spring

#### 5.2.1 Pattern 1

Obs-1 explains 20 % of the total space-time variance with a peak in southeast China (Fig. 8). Northeastward flow along the western side of an anomalous anticyclone in the tropical western Pacific transports moisture into southern China (Fig. 9).

Stephan et al. (2017a) showed that the 850 hPa circulation in Fig. 9 strongly resembles the antisymmetric 'C-mode' response to ENSO, even though the EOT timeseries itself is not significantly correlated with Niño3.4, but only with SSTs in the SCS (r=0.31, Table 3). The C-mode circulation is characterized by a strong anticyclone in the NWP, resulting from the nonlinear interaction between the annual cycle of wind and SST and ENSO variability (Stuecker et al., 2013, 2015).

All simulations show a leading EOT that resembles the observed one (Fig. 8), but with westward shifts of the EOT base

points. Simulations have weaker precipitation variability in southern China than observations, which could have important consequences for river discharge on the southern slopes of the mountainous terrain of southeast China (Fig. 1). Pattern correlations range from 0.43 (C512b) to 0.83 (A216). The pattern explains a similar fraction of variance in simulations and observations, between 16 % (C96) and 26 % (C512b).

All simulated EOTs lag ENSO (Table 3), and their circulation anomalies in the NWP show strong similarities to the observa-

tions in terms of the wind direction along the southeast coast (Fig. 9). Correlations with DJF Niño3.4 are largest in A96 (0.56) and A216 (0.66). All simulations produce the leading pattern of MAM precipitation variability for the correct physical reason.

#### 5.2.2 Pattern 2

Obs-2 is centered along the Yangtze valley, with variability of opposite phase in southeast China (Fig. 8). It explains 8 % of the total space-time variance and is associated with an upper-tropospheric wave train in the high and midlatitudes. It creates

significant upper-level divergence over the Yangtze River and convergence over the coastline of south China. Stephan et al. (2017a) showed that $Z_{200}$ anomalies are consistent with a Rossby wave pattern of Atlantic origin.

Meridional dipoles are found in A216 and all coupled simulations. C216 and C512a best match the observed pattern with pattern correlations of 0.71 (C216) and 0.87 (C512a), and similar explained variances of 9 % (C216) and 8 % (C512a). In C96 and C512b, the phase of the dipole is reversed and pattern correlations are therefore negative (-0.47 for C96 and -0.38 for

C512b). We consider those to be matches as well, as the signs of the EOT pattern and timeseries are arbitrary.

Simulations do not match Obs-2 $Z_{200}$ anomalies (not shown). C96-2 is associated with low $P_{SFC}$ and a cyclonic circulation anomaly over the coast. C512b-2 is associated with anomalous easterly flow along the Yangtze River driven by high $P_{SFC}$ over central China. A216-2, C216-2 and C512a-2 are associated with a $Z_{500}$ ridge over Japan (JPN $Z_{500}$ index in Table 3) and an anomalous anticyclonic circulation.



### 5.2.3 Pattern 3

Obs-3 rainfall anomalies are found in large areas of eastern China (Fig. 8). This pattern explains 7 % of the total space-time variance. It has a correlation of 0.36 with the preceding DJF Niño3.4 index, consistent with Fig. 4. Southeastward propagating waves from high latitudes and northeastward propagating waves from south China terminate over Korea and Japan in a region

of high pressure, indicating blocking. Stephan et al. (2017a) argue that these waves are likely triggered by anomalous equatorial heating associated with a decaying ENSO state.

Only two simulations, C216 and C512b, partly produce this pattern, with pattern correlations of 0.59 and 0.65. In these simulations anomalously dry conditions exist in southern China but not in Obs-3. In the simulations, the pattern is not associated with ENSO, but there exist significant OLR anomalies and Rossby wave sources over Africa (not shown), significant

extratropical upper-tropospheric geopotential height anomalies (not shown), and anomalously high pressure over Japan (JPN $Z_{500}$ index in Table 3), indicating that these simulations produce Obs-3 for the correct reason.

### 5.3 Summer

### 5.3.1 Pattern 1

Obs-1 explains 12 % of the total space-time variance and describes coherent precipitation variability along the southern Yangtze

valley (Fig. 10). It lags ENSO with a significant correlation of 0.38 with DJF Niño3.4, and with JJA SCS SSTs (0.42) and JJA NWP $P_{SFC}$ (0.4), respectively (Table 4). The positive NWP $P_{SFC}$ anomaly and the associated anticyclonic circulation strengthen the summer monsoon circulation, as discussed in Sect. 4.

A96, A216, C96, C216 and C512a produce similar leading patterns with pattern correlations between 0.66 (C512a) and 0.75 (A96, C96) (Fig. 10 and Table 4). The A96-1 area is larger than in observations, and the total explained variance (23 %) is

almost double the observed value. The other simulations have explained variances closer to observations between 9 % (C512a) and 15 % (C96). Only C512b does not capture this pattern.

Only C96 produces a significant correlation with DJF Niño3.4, albeit a weak one (r=0.2). This is expected because C96 is the only simulation with a significant projection of Yangtze valley rainfall onto DJF Niño3.4 (Fig. 4). In all simulations, as in observations, the pattern is accompanied by an anomalous anticyclonic circulation in the NWP (not shown). C96 and

C512a also show areas of significantly increased $P_{SFC}$ that extend far eastward into the North Pacific (Table 4). In A96, A216 and C216 this circulation is more locally confined. All GC2 simulations associate the pattern with warm SSTs in the SCS, as in observations (Table 4). This is consistent with the more accurate JJA circulation response to ENSO in GC2 than in GA6 (Fig. 5).





### 5.3.2 Pattern 2

Obs-2 peaks along the northern reaches of the Yangtze valley (Fig. 10). Precipitation variability in this region is associated with increased $P_{SFC}$ over a small area of south China that creates a regional lower tropospheric circulation anomaly with increased westerlies along the Yangtze River.

The only simulation with a pattern of coherent IAV in the northern Yangtze valley is C512b, with a pattern correlation of 0.65 with Obs-2. The pattern is C512b-3, but it explains a similar fraction of variance as Obs-2 (4 % and 5 %, respectively). Note that C512b is the only simulation that didn't capture JJA Obs-1. Pressure and circulation anomalies in C512b-3 spread further into the North Pacific than in observations (not shown).

### 5.4    Autumn

### 5.4.1    Pattern 1

Obs-1 describes rainfall anomalies in the eastern Yangtze region and an area to the southwest. The observed pattern explains 13 % of the total space-time variance (Fig. 11). It is associated with a high pressure anomaly southeast of China and southerly flow into southeast China. Significantly increased pressure is present both at the surface and in the upper troposphere (Fig 12).

The observed pattern exists in all simulations, but the southwestward extension of the area of coherent rainfall is only present

in A96, A216 and C512b. This explains the higher pattern correlations in these simulations (0.75–0.82), compared to those where variability is confined to the Yangtze (0.60–0.67). The simulations with the southwest extension explain more variance (15–30 %) than the others (7–13 %).

Fig 12 shows that in observations the high pressure anomaly over eastern China and the western North Pacific is part of a wave pattern with another positive anomaly over Europe and a negative anomaly over northwestern China and Mongolia.

Partial agreement with these anomalies is seen in all simulations except in A96. In A96 the pattern corresponding to Obs-1 is associated with ENSO; A96 has a correlation of 0.4 with the SON Niño3.4; $P_{SFC}$ and $Z_{200}$ anomalies over the eastern tropical Pacific are typical of El Niño (Fig. 12). C216-2 and C512b-1 are also weakly correlated with SON Niño3.4. The connections to ENSO in A96, C216 and C512b are consistent with Fig. 4.

### 5.4.2    Pattern 2

Obs-2 explains 9 % of the total rainfall variability in SON, with a base point located near the coast of southern China (Fig. 11). It was associated with anomalously high OLR in the western tropical Pacific, indicating suppressed convection.

A96 and all GC2 simulations also produce patterns centered in south China. They explain between 7 % (C512b) and 14 % (A96) of the space-time variance. The smallest pattern correlation of 0.68 is found in C512a, where the base point is shifted northward. In A96, EOT 2 describes a dipole between the Yangtze and the Huaihe Rivers. A216 misses the pattern.

As in the observations, the patterns in C96, C216 and C512b are associated with high OLR in the western tropical Pacific (not shown). The lack of this signature in A96 and C512a points to a different physical mechanism. In C512a the subtropical



westerly jet stream over East Asia is shifted northward and upper-level divergence over the Yangtze Basin and southern China drives a lower-tropospheric anticyclonic circulation over southeastern China. In A96 the precipitation anomaly cannot be associated with any statistically significant circulation feature.

### 5.5 Summary of EOT results

To summarize model performance for EOTs, Fig. 13 shows the patterns that each simulation was able to produce (circles) and the ones that it missed (crosses). For the former it shows the pattern correlations with the observed pattern, and the difference in the percentages of explained space-time variance. Large circles indicate that a simulated pattern is associated with a similar physical mechanism as in observations, according to the arguments presented above.

All simulations produce the leading patterns of variability in DJF, MAM and SON. The leading pattern in JJA is also present
in all simulations except for C512b. C96 and C512a capture the observed mechanisms for all leading patterns. C96 and C512a also produce the secondary patterns in MAM and SON; in C96 this includes associated mechansims. In addition, C96 produces the secondary pattern in DJF.

A96 produces six of the nine observed patterns, two of them with the observed mechanism. A216 also produces six, three of them with the observed mechanism. C96 and C512b (C512a) produce seven (eight), five (four) of them associated with the
observed mechanisms;

Thus, by these measures C216 is the best-performing simulation, especially as all patterns in C216 have pattern correlation coefficients of 0.59 or higher. One may further conclude that A96 has the poorest performance.

Comparing A96 to C96 and A216 to C216 shows that coupling increases the number of simulated patterns and the number of patterns associated with the observed physical mechanisms. In contrast, there is no evidence that simulated IAV benefits
from a higher horizontal resolution.

### 6 Discussion

In addition to examining the horizontal distributions of mean precipitation and its interannual standard deviation in Met Office GCM simulations, we performed a comprehensive assessment of the leading patterns of coherent precipitation variability in all seasons using EOT analysis. The EOT method is robust to large biases in mean precipitation. For example, A216 has the
greatest bias in IAV in DJF (Fig. 2). In A216 IAV is greatest in the eastern Yangtze valley, and not at ∼111° E, the location of the base point of the leading pattern in A216. Consistently, in Fig. 6 regressed rainfall is greatest in the eastern Yangtze River basin, not at the base point. We search for the point that best explains the area-averaged timeseries; while areas of large variability tend to contribute most to this timeseries, the base point in A216 remains close to the observed point. A similar case is in JJA, when IAV peaks in south China in all simulations, not along the Yangtze valley as is observed. Nevertheless, five of
six simulations produce a leading pattern that peaks in the southern Yangtze River valley.

We measured the similarity between simulated and observed patterns by pattern correlation coefficients and differences in explained space-time variance. These are important and objective metrics, but they are relatively insensitive to small spatial





shifts in the patterns of anomalous rainfall. For some potential uses of simulated precipitation data, such shifts may be important (e.g., hydropower); a more detailed analysis is warranted. Furthermore, we used correlations with indices of atmospheric and oceanic variability, and regressions of atmospheric fields onto EOT timeseries, to argue whether or not two patterns are associated with similar physical mechanisms. To support our classifications, we analyzed lead-lag regressions of additional

fields for observations and all simulations (including SST, OLR, surface wind, surface temperature, wind at 850 hPa, 500 hPa, 200 hPa, $Z_{500}$, $Z_{200}$, column-integrated moisture flux, divergence of horizontal wind at 200 hPa, Rossby wave sources, wave activity flux). This additional information supported the classifications we presented.

A related problem is that precipitation is inherently a local phenomenon and affected by more than one process. One example is Obs-1 in DJF, which is correlated with Niño3.4, but also shows substantial extratropical circulation anomalies that may

be unrelated to ENSO. In some simulations this pattern was correlated with Niño3.4, but in others only the extratropical circulation anomalies were statistically significant. It is possible for a simulation to fail to reproduce a significant physical mechanism in observations, but to reproduce an alternative mechanism that is not significant in observations. There may be value in simulations that produce observed patterns but not the leading mechanism.

We analyzed one GC2 and one GA6 simulation each at resolutions of N96 and N216, but two GC2 simulations at N512.

These GC2 simulations differ only in their initial conditions; both span 100 years. Their climatological mean rainfall and IAV are almost identical (Fig. 2). The teleconnection to ENSO is also similar except in SON, when Niño3.4 in C512a is negatively correlated with rainfall anomalies in central China, while C512b Niño3.4 is positively correlated with anomalies in southeast China (Fig. 4). Nevertheless, it is clear from Fig. 13 that patterns of rainfall variability and their causes are as different between C512a and C512b as they are between any other pair of simulations. This has several implications. First,

integrations over 100 years may not be long enough to eliminate effects of internal variability. Mean state biases alone cannot explain the discrepancies: the two simulations have similar biases in SST (Fig. 3), monsoon circulation, and the seasonal cycle of precipitation (not shown). The importance of internal variability was previously highlighted by Deser et al. (2012), who estimated that half of the inter-model spread in projected climate trends for air temperature, precipitation, and sea level pressure during 2005–2060 in the CMIP3 multi-model ensemble is due to internal variability. Alternatively, the observed record

used in this study is not long enough to robustly estimate precipitation variability.

A mode of internal multi-decadal variability relevant to this study is the Pacific Decadal Oscillation (PDO). ENSO affects precipitation variability in all seasons in observations and all simulations. However, the PDO modulates the effect of ENSO on East Asian precipitation during JJA, DJF and MAM (Zhang et al., 1997; Wang et al., 2008; Yang and Zhu, 2008; Wu and Mao, 2016). For instance, Wu and Mao (2016) showed when ENSO and PDO are in phase, MAM rainfall anomalies in south China

are strongly correlated with ENSO. In contrast, when ENSO and PDO are out of phase, the ENSO-precipitation relationship over China weakens or becomes insignificant. All simulations showed a significant correlation between their leading pattern of MAM precipitation variability and the preceding DJF Niño3.4, but observations did not. The absence of a significant correlation with Niño3.4 in the observations may be attributed to the PDO: the average value of the observed leading MAM EOT timeseries in units of standard deviations is 0.45 for El Niño-PDO+ (9 years), –0.43 for El Niño-PDO- (10 years), 0.00 for La Niña-PDO+





(8 years) and –0.38 for La Niña-PDO- (7 years). This example shows that modes of internal multi-decadal variability can play an important role.

To diagnose the PDO in the simulations we computed anomalies of December–May mean detrended SSTs north of 20° N in the North Pacific. We computed the first EOF of these anomalies, after weighting them by the cosine of latitude and

interpolating to a $1° \times 1°$ latitude-longitude grid. Fig. 14 shows regressions of detrended and 10-year lowpass filtered SST anomalies against the principal component timeseries for observations (1871–2010 HadISST) and the coupled simulations. The observations show typical PDO SST anomalies with opposite-signed variability in the North Pacific and along the central and eastern equatorial Pacific. In the simulations, equatorial SST anomalies are very weak. In C512, the SST pattern in the North Pacific is not correct and explains only half of the variance as in observations. This shows that the PDO in the coupled

simulations differs from the observed PDO. It is plausible that this creates biases in the ENSO-precipitation relationship over China.

## 7    Summary

This is the first study to assess GCM simulations over China in terms of their ability to reproduce not only the geographical distributions of mean rainfall and its interannual variability (IAV), but also the leading patterns of coherent precipitation vari-

ability by applying Empirical Orthogonal Teleconnection (EOT) analysis. We used EOT analysis to identify strong coherent regional rainfall variability, and regressions of atmospheric fields and SSTs onto the EOT timeseries to identify associated physical processes. Accurately simulating such variability is crucial if GCMs are to be used to understand the risk of disastrous climate impacts such as droughts and floods. We examined two climate simulations of MetUM GA6 at resolutions of N96 (A96, 200 km) and N216 (A216, 90 km), and four of GC2 at horizontal resolutions of N96 (C96, 200 km), (C216, 90 km)

and N512 (C512a and C512b, 40 km) (Table 1). For all seasons, we tested how well simulations produce observed patterns of regional precipitation variability, and whether these patterns are associated with the same physical mechanisms described in Stephan et al. (2017a) for observations.

Positive biases in simulated seasonal mean precipitation and IAV of ∼100 % are found in all seasons (Fig. 2), particularly in southern China. In most seasons and areas increasing resolution and adding air-sea coupling improve biases, particularly near

orography in southwest China. These findings are consistent with Li et al. (2015) and Gao et al. (2006), indicating that better resolved orography and more resolved physical processes may both play a role depending on season and location. The fact that the sensitivity of model performance to resolution and coupling varies seasonally and depends on the area of interest may explain the lack of consensus among previous studies on the importance of resolution.

ENSO is the main driver of IAV in China in DJF, MAM and JJA. The presence and strength of the teleconnection to ENSO

is sensitive to coupling and resolution (Fig. 4). In MetUM, coupling and resolution improve the teleconnection to ENSO in DJF and in MAM, but it remains poor in JJA in all simulations.

All simulations accurately capture the leading pattern of observed DJF precipitation variability over southeast China (Fig. 6), but only two (C216 and C512) have a significant relationship with ENSO (Table 2). Nevertheless, all simulations are similar





to observations in their associated extratropical circulations, indicating that simulated DJF precipitation variability is linked to similar physical processes as in observations (Fig. 7). In contrast, simulated DJF precipitation variability along the southeast coast (Fig. 6) is not driven by the observed mechanisms (Fig. 7).

All simulations correctly produce the leading pattern of MAM rainfall variability in southeast China in response to ENSO
(Fig. 8). The second pattern, a meridional dipole, is found in all simulations except A96, but no simulation associates it with observed anomalies (Fig. 8). The third leading pattern in MAM, located in northern China, is captured by C216 and C512b and associated with observed anomalies (Table 3).

Five simulations produce the leading pattern of JJA precipitation variability, located in the southern Yangtze valley (Fig. 10), but only C96 is weakly correlated with DJF Niño3.4. Associated observed SST anomalies in the SCS and atmospheric pressure
and circulation anomalies are present in C96 and C512a (Table 4).

All simulations except A96 capture the leading pattern of SON precipitation variability along the Yangtze River and associate it with extratropical wave disturbances similar to observations (Fig. 11). The second pattern in SON peaks in southeast China and is found in all simulations except in A216, but only C96, C216 and C512b associate it with observed processes.

Overall, coupled simulations capture more observed patterns of variability and associate more of them with the correct phys-
ical mechanism, compared to atmosphere-only simulations at the same resolution (Fig. 13). In our six simulations, changes in resolution within the range N96-N512 (200–40 km) do not change the fidelity of these patterns or their associated mechanisms. Evaluating climate models in terms of the geographical distribution of climatological mean precipitation and its interannual standard deviation is insufficient; attention should also be paid to associated mechanisms. This conclusion even holds for the two C512 simulations that differ only in their initial ocean state. Their mean rainfall and its IAV are almost identical (Fig. 2),
but patterns of rainfall variability and their causes are as different between C512a and C512b as they are between any two other simulations. Further research is needed into what causes such different model behavior, particularly when there are no apparent differences in mean state biases between these two simulations. C216 is the best-performing simulation in terms of reproducing the most patterns with the highest pattern correlation coefficients of 0.59 or higher, and in terms of reproducing the most observed mechanisms (Fig. 13 and Sect. 5.5). Similarly, one can conclude that A96 has the poorest performance.

*Code and data availability.* Data and code will be made available upon request through JASMIN (http://www.jasmin.ac.uk/).

*Competing interests.* No competing interests are present.

*Acknowledgements.* This work and its contributors (CC Stephan, PL Vidale, AG Turner, M-E Demory and L Guo) were supported by the UK-China Research & Innovation Partnership Fund through the Met Office Climate Science for Service Partnership (CSSP) China as part of the Newton Fund. NP Klingaman was supported by an Independent Research Fellowship from the Natural Environment Research Council



(NE/L010976/1). The high-resolution model C512 was developed by the JWCRP-HRCM group. The C512 simulations were supported by the NERC HPC grants FEBBRAIO and FEBBRAIO-2 (NE/R/H9/37), and they were performed on the UK National Supercomputing Service ARCHER by Prof. Pier Luigi Vidale and Karthee Sivalingam. APHRODITE data are available from http://www.chikyu.ac.jp/precip/. OLR data are provided by the NOAA/OAR/ESRL PSD, Boulder, Colorado, USA, at http://www.esrl.noaa.gov/psd/. The Rossby wave source

5   function was computed using code from the python package windspharm v1.5.0 available at http://ajdawson.github.io/windspharm.



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





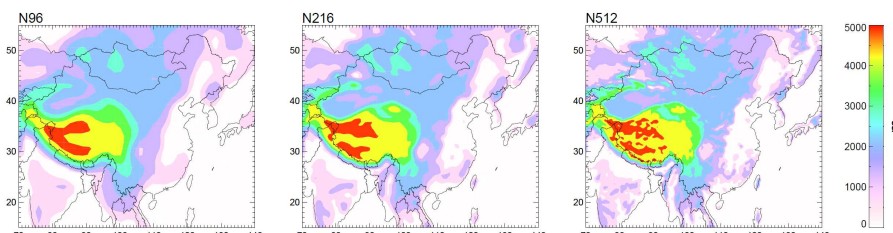

**Figure 1.** Grid-point mean orographic height for (left) N96 (middle) N216 and (right) N512 horizontal resolution.

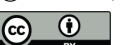



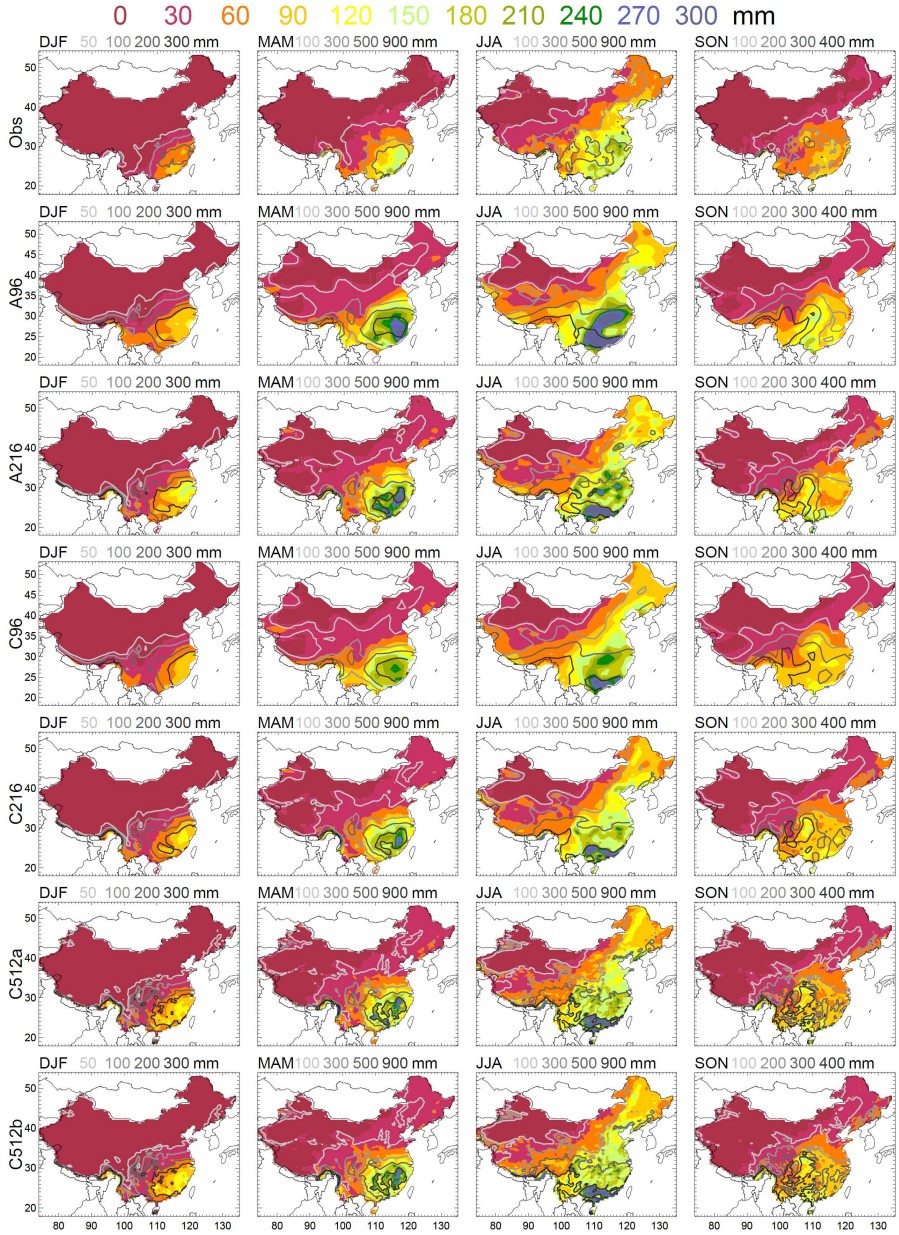

**Figure 2.** Climatological seasonal total precipitation (gray contours) and interannual standard deviation (shading) for 1951–2007 observations (top) and the full length of each simulation. The season is indicated above the panels.





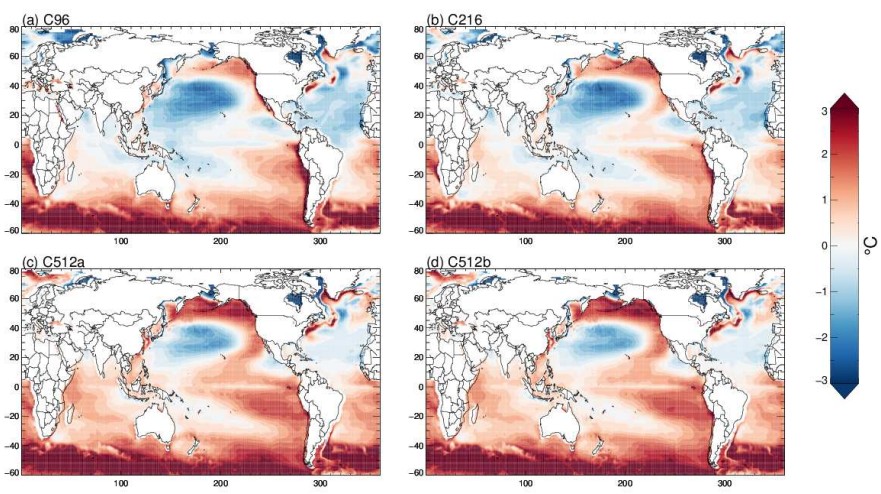

**Figure 3.** Annual mean SST bias in the coupled simulations relative to observations (1870–2010 HadISST).





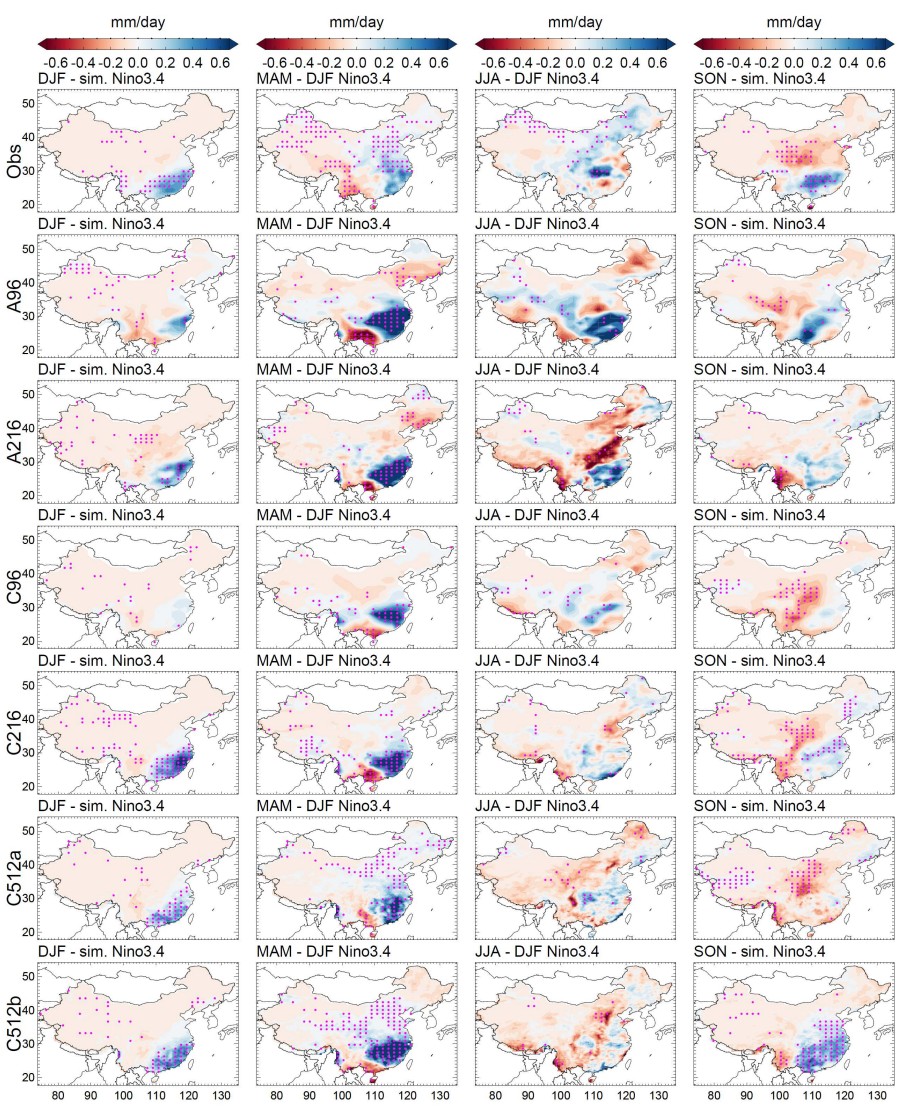

**Figure 4.** Regression of DJF precipitation against observed or simulated DJF Niño3.4 (left), lagged regression of MAM precipitation against preceding DJF Niño3.4 (middle left), lagged regression of JJA precipitation against preceding DJF Niño3.4 (middle right), regression of SON precipitation against SON Niño3.4 (right). Stippling indicates confidence levels exceeding 95 %.





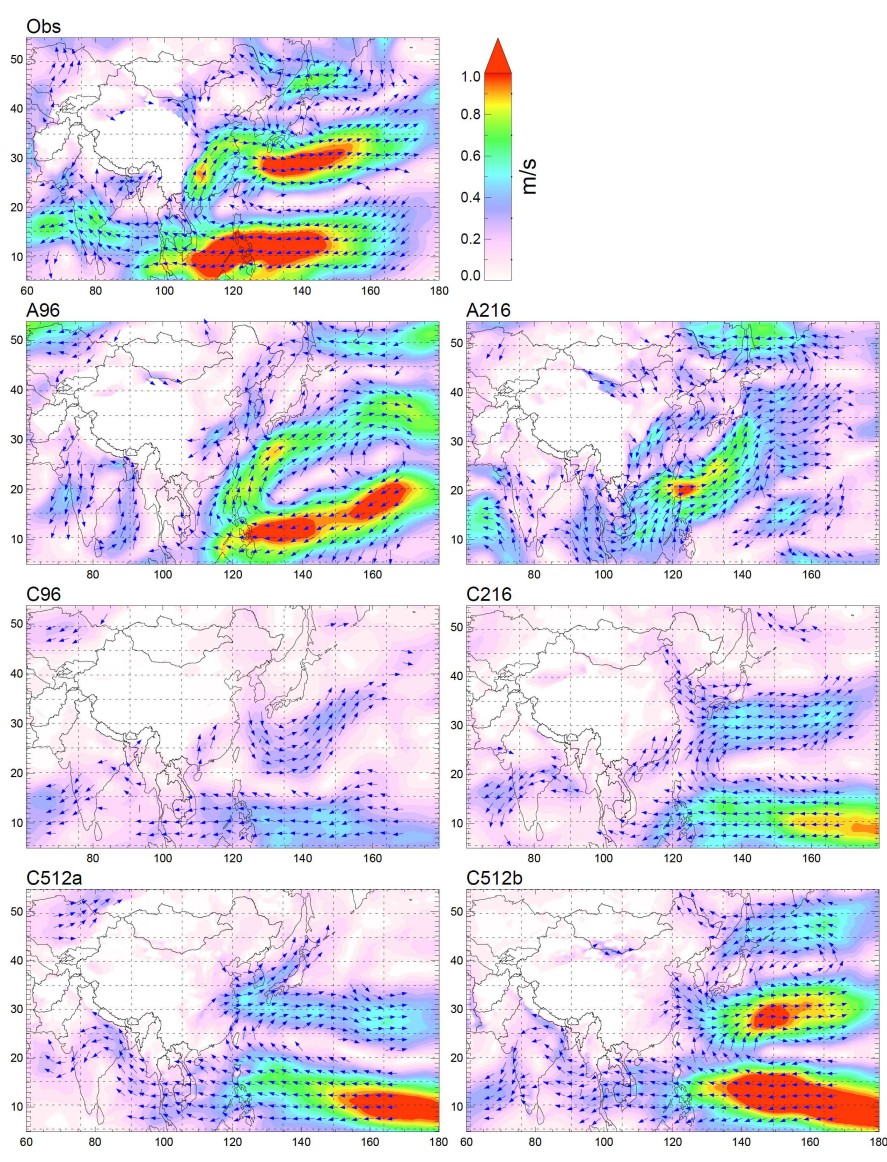

**Figure 5.** Regression of observed or simulated JJA 850 hPa wind against observed or simulated normalized DJF Niño3.4. Observations use 1982–2008 to match the years of the atmosphere-only simulations. Arrows indicate the wind direction and are drawn where wind speeds exceed $0.2 \text{ ms}^{-1}$.





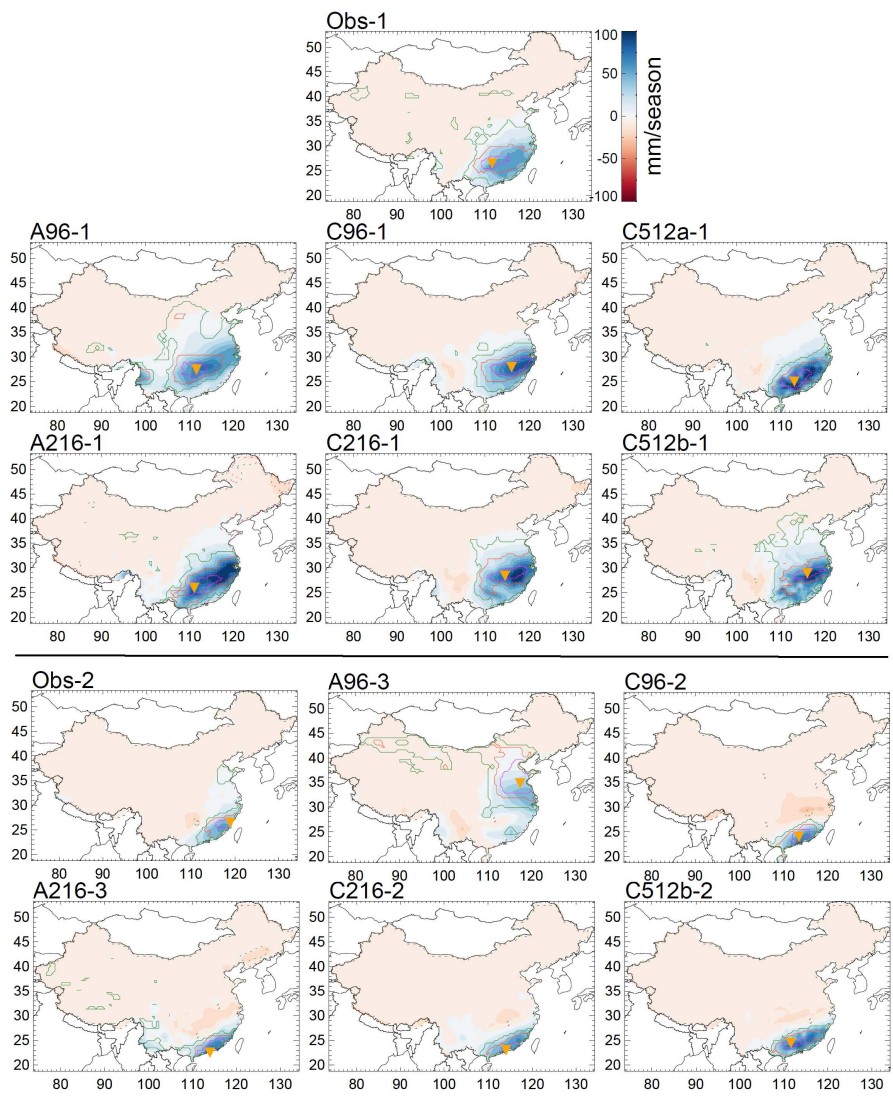

**Figure 6.** Shading shows regressions of DJF precipitation against observed or simulated EOT timeseries. Also shown are positive (solid lines) and negative (dashed lines) correlations of the full (leading order) or residual (higher order) precipitation-anomaly timeseries with the EOT base point exceeding 0.8 (magenta), 0.6 (orange), 0.4 (green). The EOT base point is marked by the orange inverted triangle. The top three (bottom two) rows show patterns that have a linear correlation coefficient exceeding 0.38 with the first (second) observed pattern. The number next to the simulation name above each panel indicates the order of the simulated EOT pattern.





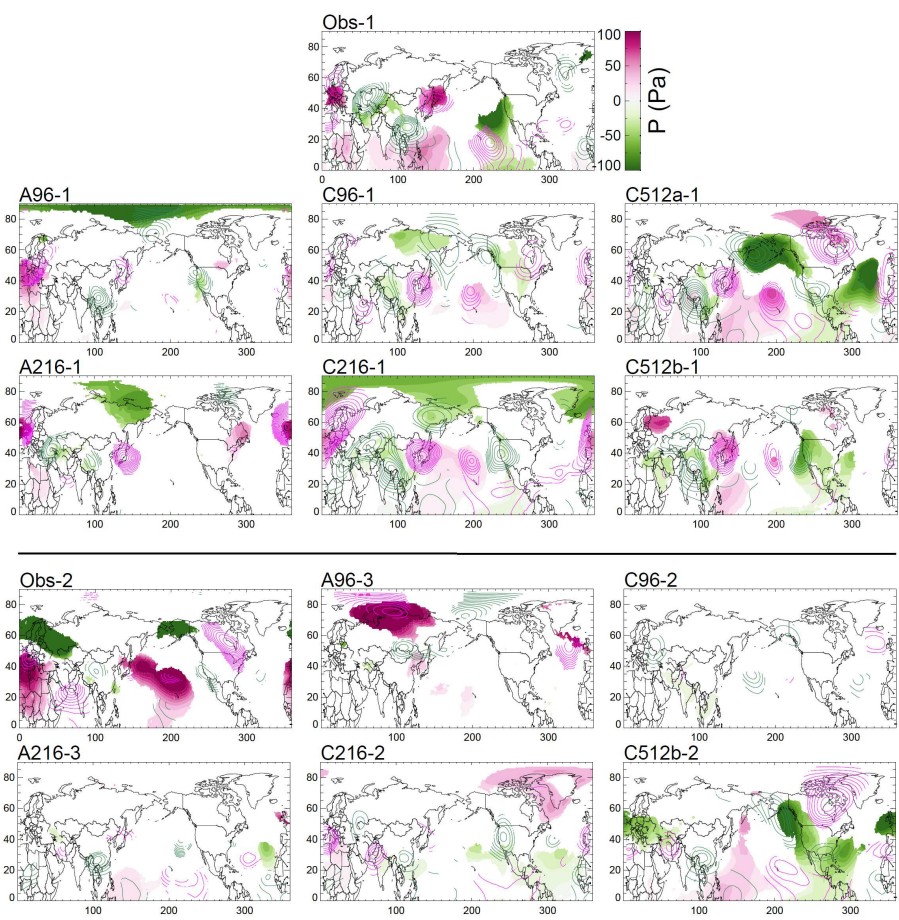

**Figure 7.** Regressions of $P_{SFC}$ (shading) and $Z_{200}$ (contours at intervals of 0.02 m, pink: positive, green: negative) against normalized DJF EOT timeseries. The top three (bottom two) rows are for EOTs matching the first (second) observed pattern. All shown values are significant at the 90 % confidence level.



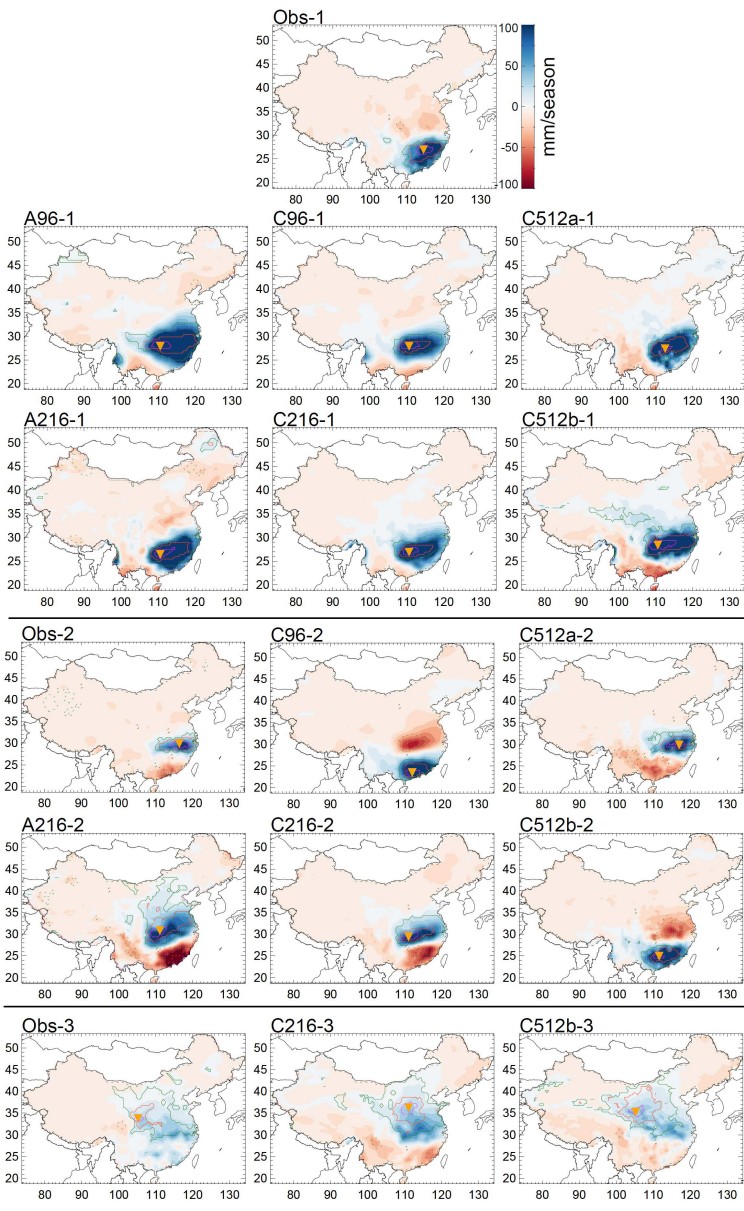

**Figure 8.** Shading shows regressions of MAM precipitation against observed or simulated EOT timeseries that have a linear correlation coefficient exceeding 0.38 with Obs-1 (top), Obs-2 (middle) or Obs-3 (bottom). Also shown are positive (solid lines) and negative (dashed lines) correlations of the full (leading order) or residual (higher order) precipitation-anomaly timeseries with the EOT base point exceeding 0.8 (magenta), 0.6 (orange), 0.4 (green). The EOT base point is marked by the orange inverted triangle. The number next to the simulation name above each panel indicates the order of the simulated EOT pattern.





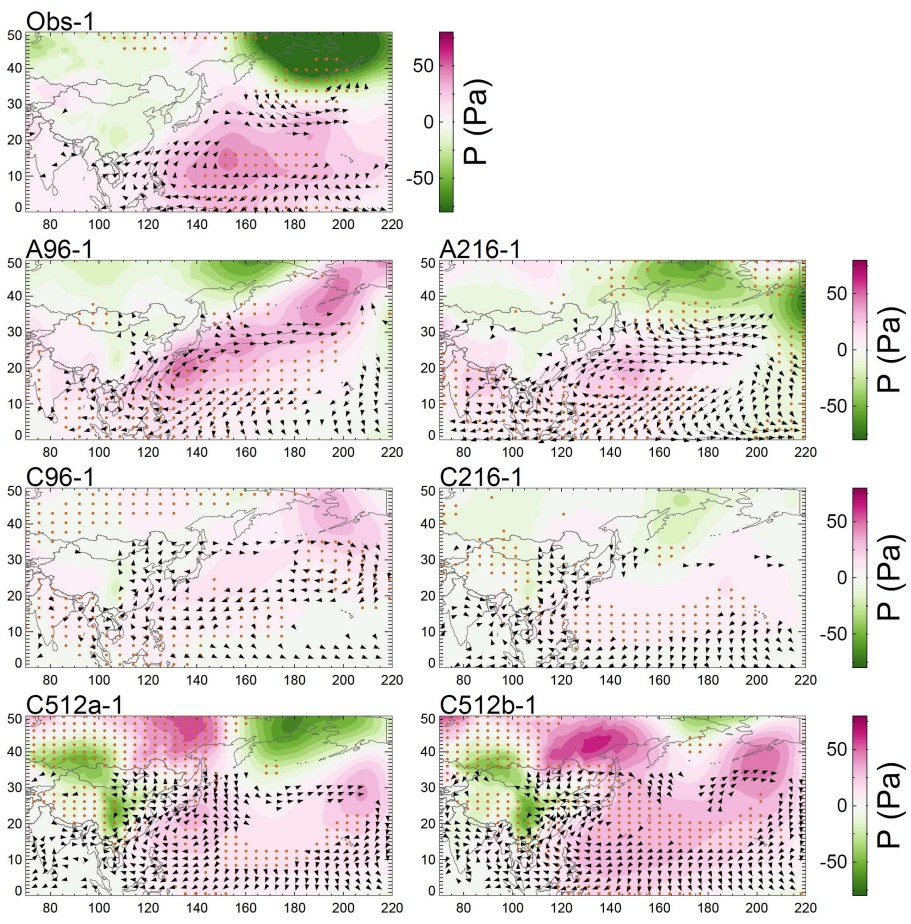

**Figure 9.** Regressions of $P_{SFC}$ (shading) and 850 hPa wind (arrows) against normalized MAM EOT timeseries corresponding to Obs-1. Stippling indicates confidence levels exceeding 90 %. Wind arrows are drawn when at least one component is statistically significant and the wind speed exceeds 0.1 ms$^{-1}$.





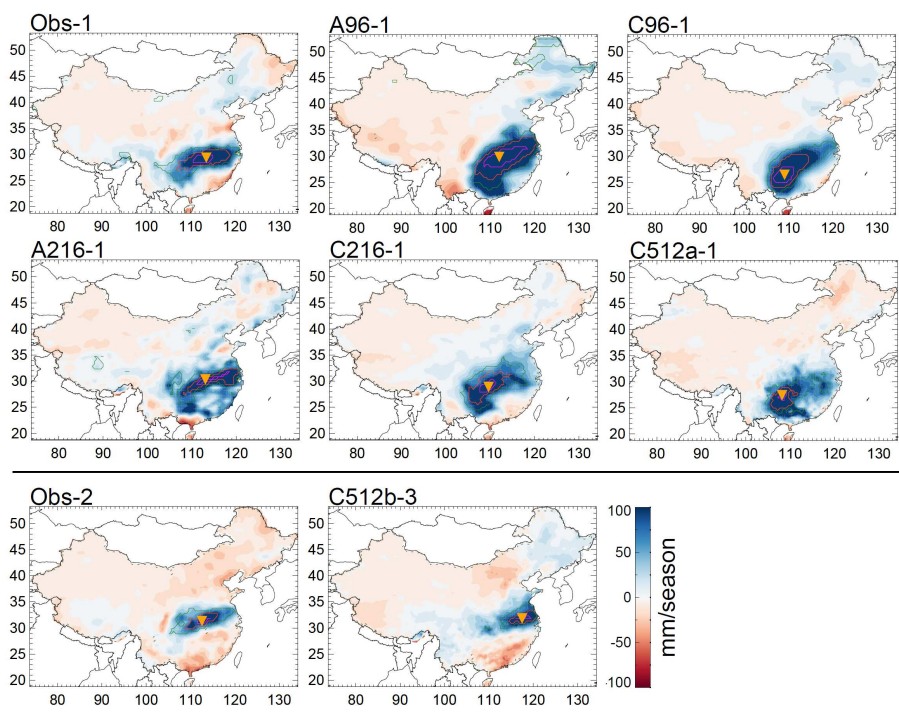

**Figure 10.** Shading shows regressions of JJA precipitation against observed or simulated EOT timeseries. Also shown are positive (solid lines) and negative (dashed lines) correlations of the full (leading order) or residual (higher order) precipitation-anomaly timeseries with the EOT base point exceeding 0.8 (magenta), 0.6 (orange), 0.4 (green). The EOT base point is marked by the orange inverted triangle. The top three (bottom) rows show patterns that have a linear correlation coefficient exceeding 0.38 with the first (second) observed pattern. The number next to the simulation name above each panel indicates the order of the simulated EOT pattern.



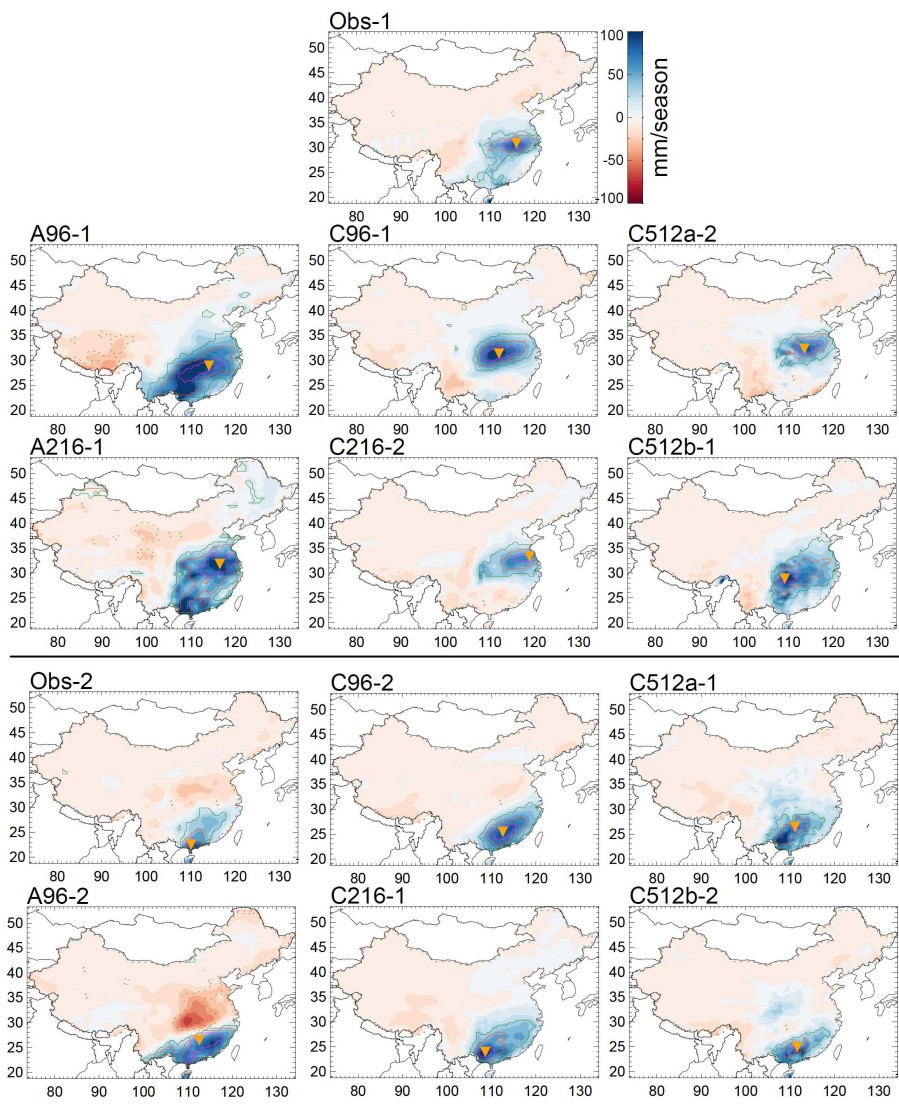

**Figure 11.** Shading shows regressions of SON precipitation against observed or simulated EOT timeseries. Also shown are positive (solid lines) and negative (dashed lines) correlations of the full (leading order) or residual (higher order) precipitation-anomaly timeseries with the EOT base point exceeding 0.8 (magenta), 0.6 (orange), 0.4 (green). The EOT base point is marked by the orange inverted triangle. The top three (bottom two) rows show patterns that have a linear correlation coefficient exceeding 0.38 with the first (second) observed pattern. The number next to the simulation name above each panel indicates the order of the simulated EOT pattern.



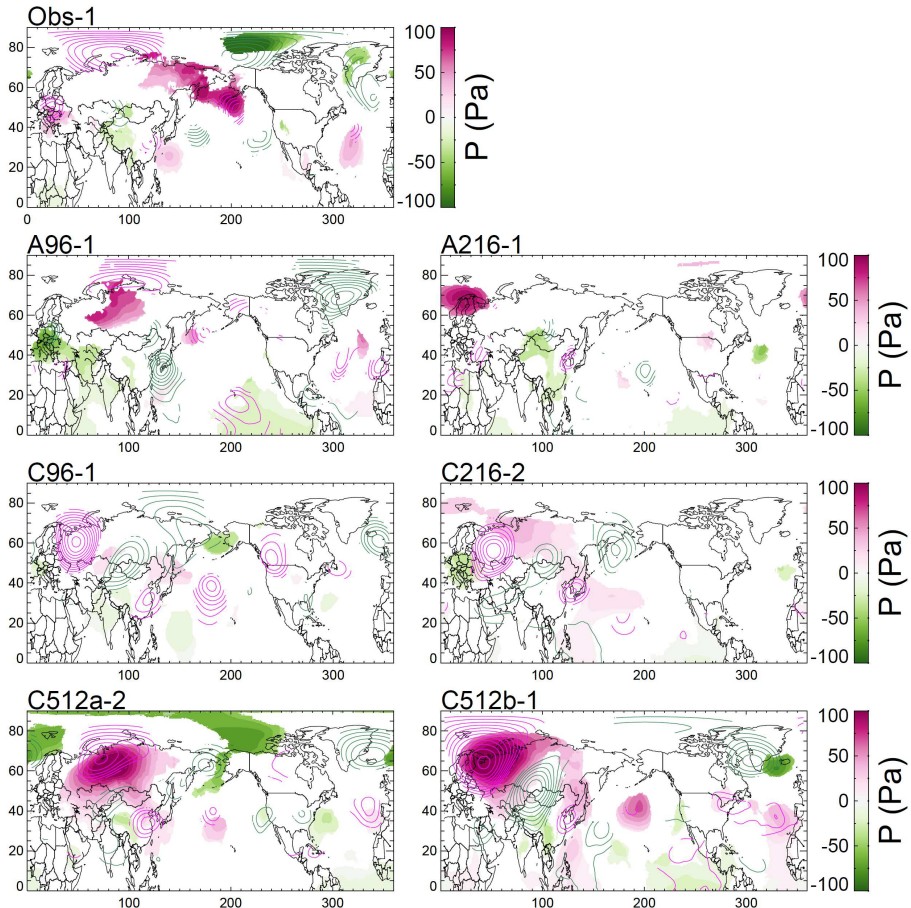

**Figure 12.** Regressions of $P_{SFC}$ (shading) and $Z_{200}$ (contours at intervals of 0.02 m, pink: positive, green: negative) against normalized SON EOT timeseries corresponding to Obs-1. All shown values are significant at the 90 % confidence level.





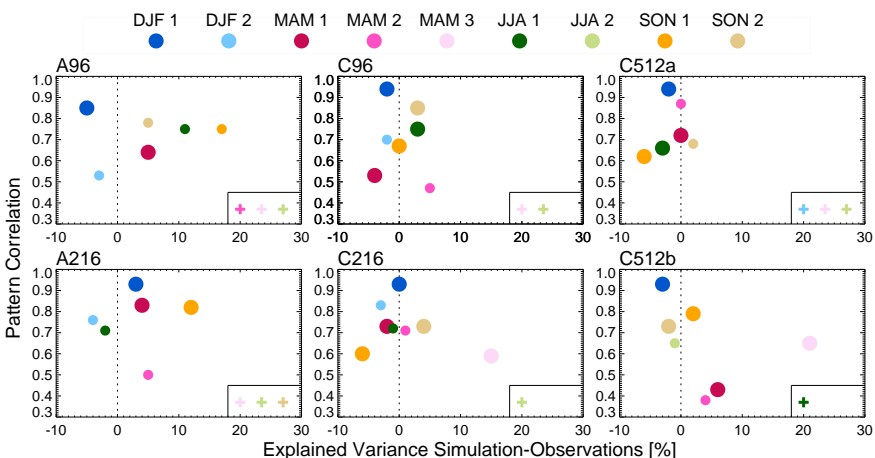

**Figure 13.** Observed patterns a simulation was able to produce (circles) and the ones that it missed (crosses). For patterns that could be reproduced, the y-axis gives the pattern correlation coefficient and the x-axis shows the difference in the percentages of explained space-time variance. Large circles indicate that a simulated pattern is associated with a similar physical mechanism as in observations.





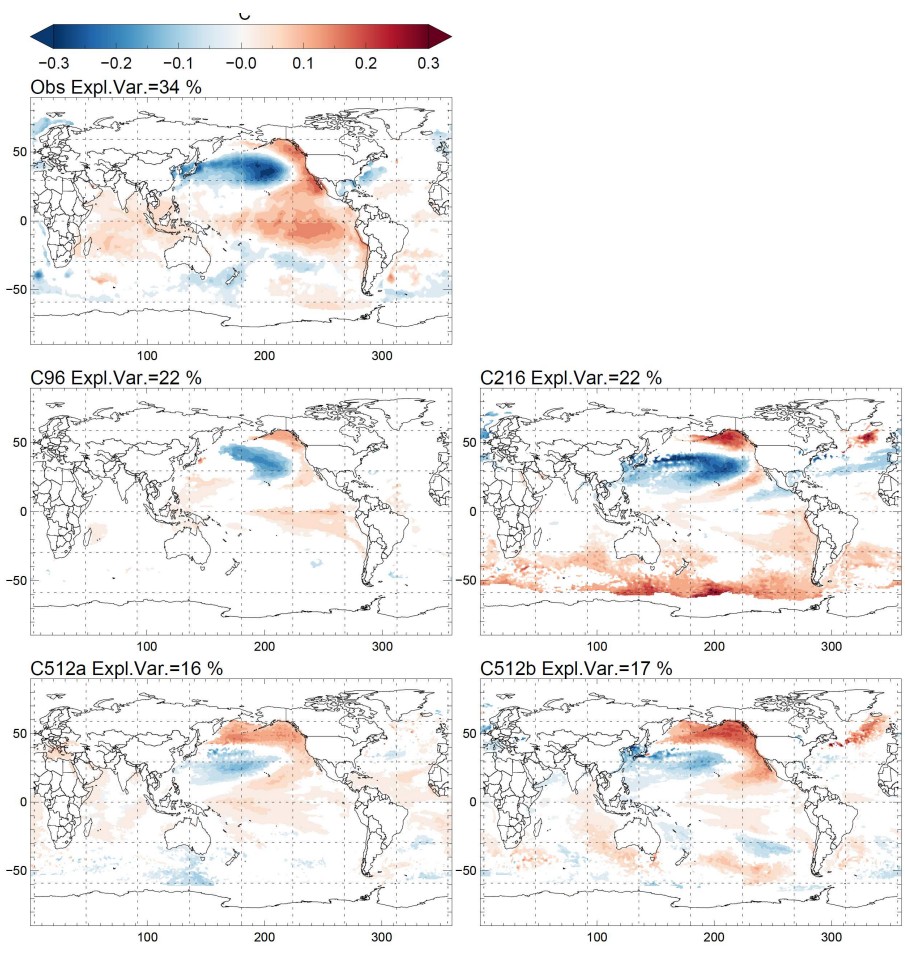

**Figure 14.** Regression against the PDO index of December–May mean detrended and lowpass-filtered (>10 years) SST anomalies for observations (1871–2010 HadISST) and the coupled simulations. All shown values exceed the 90 % confidence level. The PDO index is the principal component of the leading EOF of December–May mean detrended SST anomalies north of 20° N in the North Pacific; the explained variance is shown above the panel. See Sect. 6 for more details.



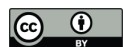

**Table 1.** The resolution, integration length, and type of ocean coupling are listed for all simulations. All simulations have 85 vertical levels with a model lid at 85 km. The resolution of the ocean is $0.25°$ for all coupled simulations.

| Simulation | Resolution | Resolution at Equator [km] | Integration length (years) | Coupling to ocean |
|:---:|:---:|:---:|:---:|:---:|
| A96 | N96 | 208 | 27 (1982–2008) | atmosphere-only |
| A216 | N216 | 88 | 27 (1982–2008) | atmosphere-only |
| C96 | N96 | 208 | 100 | coupled |
| C216 | N216 | 88 | 100 | coupled |
| C512a | N512 | 39 | 100 | coupled |
| C512b | N512 | 39 | 100 | coupled |





**Table 2.** Performance statistics of simulated EOTs compared to observations for DJF. Column (1) observed (Obs) and simulated (labeled by simulation name) EOT patterns; numbers indicate the order of the EOT pattern. (2) linear pattern correlation coefficient of simulated and observed precipitation anomalies, (3) explained space-time variance of the EOT pattern, (4) standard deviation of the EOT timeseries, (5)-(7) Spearman's rank correlation of the EOT timeseries with the (5) Niño3.4 SST index, (6) SCS SST index and (7) NWP $P_{SFC}$ index (defined in Sect. 2.3). Correlation coefficients in (5)-(7) are shown only when they are significant at the 90 % confidence level, and marked with an asterisk for confidence levels <95 %.

|  | Pattern corr. | Expl.Var.[%] | Stddev.[mm] | DJF Nino3.4 SST | DJF SCS SST | DJF NWP $P_{SFC}$ |
|---|---|---|---|---|---|---|
| Obs-1 |  | 34 | 76 | 0.44 | 0.40 | 0.42 |
| A96-1 | 0.85 | 29 | 77 | – – – – | – – – – | – – – – |
| A216-1 | 0.93 | 37 | 90 | – – – – | – – – – | – – – – |
| C96-1 | 0.94 | 32 | 92 | – – – – | 0.34 | – – – – |
| C216-1 | 0.93 | 34 | 94 | 0.42 | 0.46 | 0.27 |
| C512a-1 | 0.94 | 32 | 106 | 0.40 | 0.39 | 0.34 |
| C512b-1 | 0.93 | 31 | 104 | 0.21 | 0.32 | 0.20 |
| Obs-2 |  | 12 | 65 | – – – – | – – – – | – – – – |
| A96-3 | 0.53 | 9 | 40 | – – – – | – – – – | – – – – |
| A216-3 | 0.76 | 8 | 82 | $0.32^*$ | 0.48 | $0.31^*$ |
| C96-2 | 0.70 | 10 | 70 | – – – – | – – – – | – – – – |
| C216-2 | 0.83 | 9 | 67 | 0.24 | 0.28 | – – – – |
| C512b-2 | 0.79 | 14 | 93 | 0.29 | 0.29 | 0.26 |

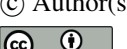



**Table 3.** Performance statistics of simulated EOTs compared to observations for MAM. Column (1) observed (Obs) and simulated (labeled by simulation name) EOT patterns; numbers indicate the order of the EOT pattern. (2) linear pattern correlation coefficient of simulated and observed precipitation anomalies, (3) explained space-time variance of the EOT pattern, (4) standard deviation of the EOT timeseries, (5)-(7) Spearman's rank correlation of the EOT timeseries with the (5) DJF Niño3.4 SST index, (6) MAM SCS SST index and (7) MAM $Z_{500}$ index over Japan (defined in Sect. 2.3). Correlation coefficients in (5)-(7) are shown only when they are significant at the 95 % confidence level.

|  | Pattern corr. | Expl.Var.[%] | Stddev.[mm] | DJF Niño3.4 | MAM SCS SST | MAM JPN $Z_{200}$ |
|---|---|---|---|---|---|---|
| Obs-1 |  | 20 | 116 | $----$ | 0.31 | $----$ |
| A96-1 | 0.64 | 25 | 192 | 0.56 | 0.44 | 0.38 |
| A216-1 | 0.83 | 24 | 211 | 0.66 | $----$ | $----$ |
| C96-1 | 0.53 | 16 | 174 | 0.38 | $----$ | 0.50 |
| C216-1 | 0.73 | 18 | 191 | 0.32 | 0.29 | $----$ |
| C512a-1 | 0.72 | 20 | 199 | 0.22 | 0.22 | 0.22 |
| C512b-1 | 0.43 | 26 | 223 | 0.39 | 0.37 | 0.42 |
| Obs-2 |  | 8 | 113 | $----$ | $----$ | $----$ |
| A216-2 | 0.50 | 13 | 83 | $----$ | $----$ | 0.68 |
| C96-2 | $-0.47$ | 13 | 163 | $----$ | $----$ | $----$ |
| C216-2 | 0.71 | 9 | 131 | $----$ | $----$ | 0.34 |
| C512a-2 | 0.87 | 8 | 223 | $----$ | $----$ | 0.20 |
| C512b-2 | $-0.38$ | 12 | 156 | $----$ | $----$ | $----$ |
| Obs-3 |  | 7 | 34 | 0.36 | $----$ | 0.52 |
| C216-3 | 0.59 | 6 | 49 | $----$ | $----$ | 0.43 |
| C512b-3 | 0.65 | 4 | 55 | $----$ | $----$ | 0.34 |



**Table 4.** Performance statistics of simulated EOTs compared to observations for JJA. Column (1) observed (Obs) and simulated (labeled by simulation name) EOT patterns; numbers indicate the order of the EOT pattern. (2) linear pattern correlation coefficient of simulated and observed precipitation anomalies, (3) explained space-time variance of the EOT pattern, (4) standard deviation of the EOT timeseries, (5)-(7) Spearman's rank correlation of the EOT timeseries with the (5) DJF Niño3.4 SST index, (6) JJA SCS SST index and (7) JJA NWP $P_{SFC}$ index (defined in Sect. 2.3). Correlation coefficients in (5)-(7) are shown only when they are significant at the 95 % confidence level.

| | Pattern corr. | Expl.Var.[%] | Stddev.[mm] | DJF Niño3.4 | JJA SCS SST | JJA NWP $P_{SFC}$ |
|---|---|---|---|---|---|---|
| Obs-1 | | 12 | 193 | 0.38 | 0.42 | 0.40 |
| A96-1 | 0.75 | 23 | 296 | – – – – | – – – – | – – – – |
| A216-1 | 0.71 | 10 | 207 | – – – – | – – – – | – – – – |
| C96-1 | 0.75 | 15 | 211 | 0.20 | 0.37 | 0.29 |
| C216-1 | 0.72 | 11 | 210 | – – – – | 0.21 | – – – – |
| C512a-1 | 0.66 | 9 | 182 | – – – – | 0.25 | 0.24 |
| Obs-2 | | 5 | 137 | – – – – | – – – – – | – – – – |
| C512b-3 | 0.65 | 4 | 155 | – – – – | – – – – | – – – – |





**Table 5.** Performance statistics of simulated EOTs compared to observations for SON. Column (1) observed (Obs) and simulated (labeled by simulation name) EOT patterns; numbers indicate the order of the EOT pattern. (2) linear pattern correlation coefficient of simulated and observed precipitation anomalies, (3) explained space-time variance of the EOT pattern, (4) standard deviation of the EOT timeseries, (5) Spearman's rank correlation of the EOT timeseries with the SON Niño3.4 that are significant at the 95 % confidence level.

| | Pattern corr. | Expl.Var.[%] | Stddev.[mm] | SON Niño3.4 |
|---|---|---|---|---|
| Obs-1 | | 13 | 100 | – – – – |
| A96-1 | 0.75 | 30 | 100 | 0.40 |
| A216-1 | 0.82 | 25 | 108 | – – – – |
| C96-1 | 0.67 | 13 | 106 | – – – – |
| C216-2 | 0.60 | 7 | 71 | 0.33 |
| C512a-2 | 0.62 | 7 | 95 | – – – – |
| C512b-1 | 0.79 | 15 | 122 | 0.27 |
| Obs-2 | | 9 | 104 | – – – – |
| A96-2 | 0.78 | 14 | 76 | – – – – |
| C96-2 | 0.85 | 12 | 97 | – – – – |
| C216-1 | 0.73 | 13 | 130 | – – – – |
| C512a-1 | 0.68 | 11 | 88 | – – – – |
| C512b-2 | 0.73 | 7 | 89 | 0.34 |