# Peer review of "Using Empirical Orthogonal Teleconnections to evaluate interannual rainfall variability over China in the Met Office Unified Model Global Atmosphere 6.0 and Global Coupled 2.0 configurations"

_Geoscientific Model Development, 2017_

## Referee Comment (RC1) · Anonymous Referee #1 · 8 Nov 2017

This paper presents a very detailed description on the interannual rainfall variability over China in the Met Office model, using a relatively new technique. The whole paper is well organized and well written. I believe it can be published as is, although I have three minor comments just so the authors are aware.

1. The title seems too long. It may be shortened by eliminating some specific information. 2. In addition to ENSO, the Indian Ocean Dipole (IOD) is another important

interannual variability, which can have an impact on rainfall over China. It would be better to elaborate a bit on IOD where appropriate. 3. Nothing political and we all know this is a scientific paper. But, we would like to include Taiwan when we say "China".

---

## Short Comment (SC1) · 13 Nov 2017

Claudia

As explained in https://www.geoscientific-model-development.net/about/manuscript_types.html GMD is encouraging that authors upload the program code of models (including relevant data sets) as a supplement or make the code and data available at a data repository preferable with an associated DOI (digital object identifier) for the exact

model version described in the paper. If for some reason the code and/or data cannot be made available in this form authors need to state the reasons for why access is restricted.

Can you please clarify how exactly the code is accessible? At this point access through JASMIN is not available and it is not clear if the access will be persistent.

All the best Lutz Gross GMD Executive Editor

---

## Short Comment (SC2) · 24 Nov 2017

Is this short comment about the MetUM code? The code is stored in a central Met Office repository, restricted by license. We cannot grant anyone access to the Met Office's code. Uploading the model output would require enormous amounts of disk space. Please advise us of further action required on our part.

Kind regards, Claudia Stephan

---

## Referee Comment (RC2) · Anonymous Referee #2 · 31 Jan 2018

This manuscript extends the observational analysis of Stephan et al. (2017a) using Empirical Orthogonal Teleconnections to examine interannual rainfall variability over China as produced by a model (both in atmosphere-only and coupled mode) at varying horizontal grid spacing. The authors conclude, based on their results, that coupling the atmosphere to the ocean produces improved interannual variability in precipitation over China, while changes in grid spacing show no consistent response. The metric for deciding this conclusion is the number of seasonal patterns produced by the model that

have large-scale meteorological conditions matching the observations. Unfortunately, examining the effect of coupling the system for any individual season does not show a consistent response. Ultimately, the result that the precipitation variability is insensitive to horizontal resolution will help guide simulation choices for the future, making this work a useful contribution to the field.

While I appreciate the authors' very thorough analysis, I found it difficult to follow the discussion of the figures at times owing to the small panel sizes and inconsistent layout of which models were presented (only those with statistically significant responses were shown forcing the selection of models to change from figure to figure, and from EOT pattern to EOT pattern). Perhaps since much of the discussion is based around how the model deviates from observations, difference plots would be more informative than plotting the mean and interannual variability (such as in Fig. 2). Maybe some of the other plots could be simplified somehow?

I have a few other minor comments listed below.

What grid spacing is the model tuning done for? Since the authors suggest C216 performs the best, it would be of interest to know whether this is because the parameterization constants have been optimized for this configuration or not.

Page 6, lines 23-24: Is the "drastic improvement" going from C96 to C216? Also, when it is mentioned the improvement is seen over the South China Sea, is this over ocean only? If so, why are the ocean points not shown in the corresponding figure?

Page 15, line 15: the correlation coefficient is 0.44, "so ENSO explains only ~40% of the variance..." Typically explained variance is the square of the correlation coefficient. Can you explain how you compute the variance explained if not from the correlation coefficient here?

Page 13, line 14: "C96 and C512b (C512a) produce seven (eight), five (four) of [the observed patterns] associated with the observed mechanisms;" Figure 13 shows C512a

produces six, not eight, of observed patterns. Also, C96 and C512b do not both produce seven of the observed patterns (as shown in Fig. 13). This line need to be clarified. Additionally, the sentence ends with a semi-colon as written, was there meant to be more there?

———————————————————

---

## Author Comment (AC1) · 1 Mar 2018

We thank the two reviewers for their helpful comments. In the following we list the changes and additions that we intend to make. We kindly ask the Editorial Office to let us know about the possibility to publish supplemental figures and to suggest the best way to do so.

Reviewer #1

[Figure]

This paper presents a very detailed description on the interannual rainfall variability over China in the Met Office model, using a relatively new technique. The whole paper is well organized and well written. I believe it can be published as is, although I have three minor comments just so the authors are aware.

1. The title seems too long. It may be shortened by eliminating some specific information.

We agree with this and will change the title to 'Interannual rainfall variability over China in the MetUM GA6 and GC2 configurations'

2. In addition to ENSO, the Indian Ocean Dipole (IOD) is another important interannual variability, which can have an impact on rainfall over China. It would be better to elaborate a bit on IOD where appropriate.

Our paper uses the results obtained in Stephan et al., 2017 as the basis for the model evaluation. Stephan et al., 2017 considered the IOD as a potential physical driver of precipitation variability but did not find evidence that the IOD was connected to any specific EOT pattern. We will add this information to this paper.

3. Nothing political and we all know this is a scientific paper. But, we would like to include Taiwan when we say "China".

We understand the reviewer's wish that Taiwan should be included. However, we have good scientific reasons why we cannot change this aspect of our analysis: the observational analysis by Stephan et al., 2017 did not include Taiwan and including Taiwan would change the EOT patterns.

Reviewer #2

This manuscript extends the observational analysis of Stephan et al. (2017a) using Empirical Orthogonal Teleconnections to examine interannual rainfall variability over China as produced by a model (both in atmosphere-only and coupled mode) at varying horizontal grid spacing. The authors conclude, based on their results, that coupling

the atmosphere to the ocean produces improved interannual variability in precipitation over China, while changes in grid spacing show no consistent response. The metric for deciding this conclusion is the number of seasonal patterns produced by the model that have large-scale meteorological conditions matching the observations. Unfortunately, examining the effect of coupling the system for any individual season does not show a consistent response. Ultimately, the result that the precipitation variability is insensitive to horizontal resolution will help guide simulation choices for the future, making this work a useful contribution to the field. While I appreciate the authors' very thorough analysis, I found it difficult to follow the discussion of the figures at times owing to the small panel sizes and inconsistent layout of which models were presented (only those with statistically significant responses were shown forcing the selection of models to change from figure to figure, and from EOT pattern to EOT pattern). Perhaps since much of the discussion is based around how the model deviates from observations, difference plots would be more informative than plotting the mean and interannual variability (such as in Fig. 2). Maybe some of the other plots could be simplified somehow?

We understand that changing the layout between figures is not ideal. However, we calculated that creating figures with identical layout would require us to add 6 figures to our manuscript. This does not seem reasonable. If the Editorial Office allows it, then we would like to add a set of figures with a consistent layout as supplemental material. Similarly, we would like to leave Figure 2 as it is, but could offer to add the same figure as a difference plot as supplemental material. Showing absolute values in Figure 2 is more appropriate because the EOT analysis identifies patterns in areas of large mean precipitation and precipitation variability. These cannot be seen in difference plots.

I have a few other minor comments listed below. What grid spacing is the model tuning done for? Since the authors suggest C216 performs the best, it would be of interest to know whether this is because the parameterization constants have been optimized for this configuration or not. The tuning is normally done at a resolution of N96 and most parameterization constants do not change when changing the resolution. We

will add this information to the manuscript with reference to the papers describing the parameter choices in the model.

Page 6, lines 23-24: Is the "drastic improvement" going from C96 to C216? Also, when it is mentioned the improvement is seen over the South China Sea, is this over ocean only? If so, why are the ocean points not shown in the corresponding figure? We admit that these sentences are not clear and do not convey useful information. We intend to delete them.

Page 15, line 15: the correlation coefficient is 0.44, "so ENSO explains only ∼40% of the variance..." Typically explained variance is the square of the correlation coefficient. Can you explain how you compute the variance explained if not from the correlation coefficient here?

We thank the reviewer for catching this. We will replace ∼40% with ∼20%.

Page 13, line 14: "C96 and C512b (C512a) produce seven (eight), five (four) of [the observed patterns] associated with the observed mechanisms;" Figure 13 shows C512a produces six, not eight, of observed patterns. Also, C96 and C512b do not both produce seven of the observed patterns (as shown in Fig. 13). This line need to be clarified. Additionally, the sentence ends with a semi-colon as written, was there meant to be more there?

We thank the reviewer for pointing this out. One circle in Fig. 13 was also covered by another circle. We will replace Fig. 13 with the correct version and correct the numbers in the text. The paragraph is missing a sentence detailing the patterns that were produced in C216. This will be added as well.

───────────────────────

---

## Author Response (AR1)

**Response to Reviewers' Comments**

We thank the two reviewers for their helpful comments. In the following we list the changes and additions that we made in order to improve our manuscript.

**Reviewer #1**

This paper presents a very detailed description on the interannual rainfall variability over China in the Met Office model, using a relatively new technique. The whole paper is well organized and well written. I believe it can be published as is, although I have three minor comments just so the authors are aware.

1. The title seems too long. It may be shortened by eliminating some specific information.

*We agree with this and changed the title to 'Interannual rainfall variability over China in the MetUM GA6 and GC2 configurations'*

2. In addition to ENSO, the Indian Ocean Dipole (IOD) is another important interannual variability, which can have an impact on rainfall over China. It would be better to elaborate a bit on IOD where appropriate.

*It is certainly true that the IOD is important for interannual precipitation variability in specific regions of China. Our paper uses the results obtained in Stephan et al., 2017 as the basis for the model evaluation. Stephan et al., 2017 considered the IOD as a potential driver of precipitation variability, but did not find evidence that the IOD was connected to any specific EOT pattern. To point this out, we added the following sentences on page 7 at lines 10–12 which include two references to papers that studied the effect of the IOD on precipitation over China:*
*"While SST variability in other ocean basins has also been connected to IAV in China (e.g., Qiu et al., 2014, Cao et al., 2014), Stephan et al. (2017a) did not find an influence of other basins on their EOT patterns. Therefore, we here focus on the teleconnection to ENSO."*

3. Nothing political and we all know this is a scientific paper. But, we would like to include Taiwan when we say "China".

*We understand the reviewer's wish that Taiwan should be included. However, we have good scientific reasons why we cannot change this aspect of our analysis: the observational EOT analysis by Stephan et al. (2017) did not include Taiwan. Including Taiwan would change the observed EOT patterns.*

**Reviewer #2**

This manuscript extends the observational analysis of Stephan et al. (2017a) using Empirical Orthogonal Teleconnections to examine interannual rainfall variability over China as produced by a model (both in atmosphere-only and coupled mode) at varying horizontal grid spacing. The authors conclude, based on their results, that coupling the atmosphere to the ocean produces improved interannual variability in precipitation over China, while changes in grid spacing show no consistent response. The metric for deciding this conclusion is the number

of seasonal patterns produced by the model that have large-scale meteorological conditions matching the observations. Unfortunately,
examining the effect of coupling the system for any individual season does not show a consistent response. Ultimately, the result that the precipitation variability is insensitive to horizontal resolution will help guide simulation choices for the future, making this work a useful contribution to the field.
While I appreciate the authors' very thorough analysis, I found it difficult to follow the discussion of the figures at times owing to the small panel sizes and inconsistent layout of which models were presented (only those with statistically significant responses were shown forcing the selection of models to change from figure to figure, and from EOT pattern to EOT pattern). Perhaps since much of the discussion is based around how the model deviates from observations, difference plots would be more informative than plotting the mean and interannual variability (such as in Fig. 2). Maybe some of
the other plots could be simplified somehow?

*We understand that changing the layout between figures is not ideal. However, we calculated that creating figures with identical layout would require us to add 6 figures to our manuscript. This does not seem reasonable. We have included the full set of figures using an identical layout at the bottom of this response. It is clear that they would take up too much space in the journal.Including them here allows them to be archived by GMD as part of the discussion manuscript.*

*We appreciate the idea to change Figure 2 into a difference plot, but decided to keep Figure 2 as it is, i.e., showing absolute values. Showing absolute values in Figure 2 is more appropriate because the EOT analysis identifies patterns in areas of large mean precipitation and precipitation variability. These cannot be seen in difference plots. The corresponding difference plot is also included at the end of this document (last figure).*

**I have a few other minor comments listed below.**

What grid spacing is the model tuning done for? Since the authors suggest C216 performs the best, it would be of interest to know whether this is because the parameterization constants have been optimized for this configuration or not.

*Model development and evaluation is typically performed at N96 resolution; most parameterization constants do not change when changing the resolution. We added the following sentence on page 4 at lines 21–22:*
*"Please refer to Walters et al. (2017) for further details on physical parameterizations and their dependence on resolution."*

Page 6, lines 23-24: Is the "drastic improvement" going from C96 to C216? Also, when it is mentioned the improvement is seen over the South China Sea, is this over ocean only? If so, why are the ocean points not shown in the corresponding figure?

*We admit that these sentences are not clear and do not convey useful information. Therefore, we deleted them.*

Page 15, line 15: the correlation coefficient is 0.44, "so ENSO explains only ~40% of the variance..." Typically explained variance is the square of the correlation coefficient. Can you explain how you compute the variance explained if not from the correlation coefficient here?

*We thank the reviewer for catching this. We replaced ~40 % with ~20 % on page 9 at line 15.*

Page 13, line 14: "C96 and C512b (C512a) produce seven (eight), five (four) of [the observed patterns] associated with the observed mechanisms;" Figure 13 shows C512a produces six, not eight, of observed patterns. Also, C96 and C512b do not both produce seven of the observed patterns (as shown in Fig. 13). This line need to be clarified. Additionally, the sentence ends with a semi-colon as written, was there meant to be more there?

*We thank the reviewer for paying attention to detail and for pointing this out. One circle in Fig. 13 was also covered by another circle. We replaced Fig. 13 with the corrected version and corrected the numbers in the text. The paragraph was missing a sentence detailing the patterns that were produced in C216. This was added as well. These changes can be found on page 13 at lines 16–17.*

**Other Changes:**

One additional change we made is to remove Table 1, which summarized the simulations. All information contained in this table could already be found it the text. Therefore, no information I lost. The reason for the removal is that we included a similar table in a different publication, which is currently in press at the Journal of Advances in Atmospheric Science. Their Editorial Board asked us to remove the Table from this paper in order not to duplicate published material.

**Supplemental Figures:**

In the following we show the set of figures that would need to be used to replace the figures in the current manuscript if we used the same layout for the same type of plot. Doing so would add 6 plots to the manuscript and we therefore decided to leave the figures as they are.

[revised manuscript text omitted]

Regressions of DJF PSFC (shading) and Z200 (contours at intervals of 0.02 m, pink: positive, green: negative) against normalized EOT timeseries corresponding to observed EOT-1. All shown values are significant at the 90 % confidence level.

[Figure]

Regressions of DJF PSFC (shading) and Z200 (contours at intervals of 0.02 m, pink: positive, green: negative) against normalized EOT timeseries corresponding to observed EOT-2. All shown values are significant at the 90 % confidence level.

[Figure]

Difference of the interannual standard deviation of precipitation between the full length of each simulation and 1951--2007 observations. The season is indicated above the panels.